# META-REINFORCEMENT LEARNING WITH INFORMED POLICY REGULARIZATION

## ABSTRACT

Meta-reinforcement learning aims at finding a policy able to generalize to new environments. When facing a new environment, this policy must explore to identify its particular characteristics and then exploit this information for collecting reward. We consider the *online adaptation* setting where the agent needs to trade-off between the two types of behaviour within the same episode. Even though policies based on recurrent neural networks can be used in this setting by training them on multiple environments, they often fail to model this trade-off, or solve it at a very high computational cost. In this paper, we propose a new algorithm that uses privileged information in the form of a task descriptor at train time to improve the learning of recurrent policies. Our method learns an informed policy (i.e., a policy receiving as input the description of the current task) that is used to both construct task embeddings from the descriptors, and to regularize the training of the recurrent policy through parameters sharing and an auxiliary objective. This approach significantly reduces the learning sample complexity without altering the representational power of RNNs, by focusing on the relevant characteristics of the task, and by exploiting them efficiently. We evaluate our algorithm in a variety of environments that require sophisticated exploration/exploitation strategies and show that it outperforms vanilla RNNs, Thompson sampling and the task-inference approaches to meta-reinforcement learning.

## 1 INTRODUCTION

Deep Reinforcement Learning has been used to successfully train agents on a range of challenging environments such as Atari games (Mnih et al., 2013; Bellemare et al., 2013; Hessel et al., 2017) or continuous control (Peng et al., 2017; Schulman et al., 2017). Nonetheless, in these problems, RL agents perform exploration strategies to discover the environment and implement algorithms to learn a policy that is tailored to solving a *single task*. Whenever the task changes, RL agents generalize poorly and the whole process of exploration and learning restarts from scratch. On the other hand, we expect an intelligent agent to fully master a *problem* when it is able to generalize from a few instances (tasks) and achieve the objective of the problem under many variations of the environment. For instance, children *know* how to ride a bike (i.e., the problem) when they can reach their destination irrespective of the specific bike they are riding, which requires to adapt to the weight of the bike, the friction of the brakes and tires, and the road conditions (i.e., the tasks).

How to enable agents to generalize across tasks has been studied in *Multi-task Reinforcement Learning* (e.g. Wilson et al., 2007; Teh et al., 2017), *Transfer Learning* (e.g. Taylor & Stone, 2011; Lazaric, 2012) and *Meta-Reinforcement Learning* (Finn et al., 2017; Hausman et al., 2018; Rakelly et al., 2019; Humplik et al., 2019). These works fall into two categories. *Learning to learn* approaches aim at speeding up learning on new tasks, by pre-training feature extractors or learning good initializations of policy weights (Raghu et al., 2019).

In contrast, we study in this paper the *online adaptation* setting where a single policy is trained for a fixed family of tasks. When facing a new task, the policy must then balance *exploration* (or probing), to reduce the uncertainty about the current task, and *exploitation* to maximize the cumulative reward of the task. Agents are evaluated on their ability to manage this trade-off within a **single episode** of the same task. The online adaptation setting is a special case of a partially observed markov decision problem, where the unobserved variables are the descriptors of the current task. It is thus

Figure 1: An environment with two tasks: the goal location ($G1$ or $G2$) changes at each episode. The sign reveals the location of the goal. Optimal informed policies are shortest paths from *start* to either $G1$ or $G2$, which never visit the sign. Thompson sampling cannot represent the optimal exploration/exploitation policy (go to the sign first) since going to the sign is not feasible by any informed policy.

possible to rely on recurrent neural networks (RNNs) (Bakker, 2001; Heess et al., 2015), since they can theoretically represent optimal policies in POMDPs if given enough capacity. Unfortunately, the training of RNN policies has often prohibitive sample complexity and it may converge to suboptimal local minima.

To overcome this drawback, efficient online adaptation methods leverage the knowledge of the task at training time. The main approach is to pair an exploration strategy with the training of *informed policies*, i.e. policies taking the description of the current task as input. *Probe-then-Exploit* (PTE) algorithms (e.g. Zhou et al., 2019) operate in two stages. They first rely on an exploration policy to identify the task. Then, they commit to the identified task by playing the associated informed policy. *Thompson Sampling* (TS) approaches (Thompson, 1933; Osband et al., 2016; 2019) maintain a distribution over plausible tasks and play the informed policy of a task sampled from the posterior following a predefined schedule. PTE and TS are expected to be sample-efficient relatively to RNNs as learning informed policies is a fully observable problem. However, as we discuss in Section 3, PTE and TS cannot represent effective exploration/exploitation policies in many environments. Humplik et al. (2019) proposed an alternative approach, *Task Inference* (TI), which trains a full RNN policy with the current task prediction as an auxiliary loss. TI avoids the suboptimality of PTE/TS by not constraining the structure of the exploration/exploitation policy. However, in TI, the task descriptors are used as targets and not as inputs, so TI focuses on reconstructing even irrelevant features of the task descriptor and it does not leverage the faster learning of informed policies.

In this paper, we introduce IMPORT (*InforMed POlicy RegularizaTion*), a novel policy architecture for efficient online adaptation that combines the rich expressivity of RNNs with the efficient learning of informed policies. At train time, a shared policy head receives as input the current observation, together with either a (learned) embedding of the current task, or the hidden state of an RNN such that the informed policy and the RNN policy are learned simultaneously. At test time, the hidden state of the RNN replaces the task embedding, and the agent acts without having access to the current task. This leads to several advantages: **1)** IMPORT benefits from informed policy to speed up learning; **2)** it avoids to reconstruct features of the task descriptor that are irrelevant for learning; and as a consequence, **3)** it adapts faster to unknown environments, showing better generalization capabilities.

We evaluate IMPORT against the main approaches to online adaptation on environments that require sophisticated exploration/exploitation strategies. We confirm that TS suffers from its limited expressivity, and show that the policy regularization of IMPORT significantly speeds up learning compared to TI. Moreover, the learnt task embeddings of IMPORT make it robust to irrelevant or minimally informative task descriptors, and able to generalize when learning on few training tasks.

## 2 SETTING

Let $\mathcal{M}$ be the space of possible tasks. Each $\mu \in \mathcal{M}$ is associated to an episodic $\mu$-MDP $M_\mu = (\mathcal{S}, \mathcal{A}, p^\mu, r^\mu, \gamma)$ whose dynamics $p^\mu$ and rewards $r^\mu$ are task dependent, while state and action spaces are shared across tasks and $\gamma$ is the discount factor. The descriptor $\mu$ can be a simple id ($\mu \in \mathbb{N}$) or a set of parameters ($\mu \in \mathbb{R}^d$).

When the reward function and the transition probabilities are unknown, RL agents need to devise a strategy that balances exploration to gather information about the system and exploitation to maximize the cumulative reward. Such a strategy can be defined as the solution of a partially observable MDP (POMDP), where the hidden variable is the descriptor $\mu$ of the MDP. Given a trajectory $\tau_t = (s_1, a_1, r_1, \ldots, s_{t-1}, a_{t-1}, r_{t-1}, s_t)$, a POMDP policy $\pi(a_t|\tau_t)$ maps the trajectory to actions. In particular, the optimal policy in a POMDP is a history-dependent policy that uses $\tau_t$ to construct a belief state $b_t$, which describes the uncertainty about the task at hand, and then maps it to the action that maximizes the expected sum of rewards (e.g. Kaelbling et al., 1998). In this case,

maximizing the rewards may require taking explorative actions that improve the belief state enough so that future actions are more effective in collecting reward.

The task is sampled at the beginning of an episode from a distribution $q(\mu)$. After training, the agent returns a policy $\pi(a_t|\tau_t)$ that aims at maximizing the average performance across tasks generated from $q$, i.e.,

$$\mathbb{E}_{\mu \sim q(\mu)}\left[\sum_{t=1}^{|\tau|} \gamma^{t-1} r_t^\mu \,\bigg|\, \pi\right]. \tag{1}$$

where the expectation is taken over a full-episode trajectory $\tau$ and task distribution $q$, and $|\tau|$ is the length of the trajectory. The objective is then to find an architecture for $\pi$ that is able to express strategies that perform the best according to Eq. 1 and, at the same time, can be efficiently learned even for moderately short training phases.

At training time, we assume the agent has unrestricted access to the task descriptor $\mu$. Access to such a task descriptor during training is a common assumption in the multi-task literature and captures a large variety of concrete problems. It can be of two types: i) a vector of features corresponding to (physical) parameters of the environment/agent (for instance, such features maybe available in robotics, or when learning on a simulator) (Yu et al., 2018; Mehta et al., 2019; Tobin et al., 2017). ii) It can be a single task identifier (i.e an integer) which is a less restrictive assumption (Choi et al., 2001; Humplik et al., 2019) and corresponds to different concrete problems: learning in a set of $M$ training levels in a video game, learning to control $M$ different robots or learning to interact with $M$ different users.

## 3 RELATED WORK AND CONTRIBUTIONS

In this section, we review how the online adaptation setting has been tackled in the literature. The main approaches are depicted in Fig. 2. We first compare the different methods in terms of expressiveness, and whether they leverage the efficient learning of informed policies. We then discuss learning task embeddings and how the various methods deal with unknown or irrelevant task descriptors. The last subsection summarizes our contributions.

**Evaluation of RL agent in Meta-Reinforcement Learning.** The *online adaptation* evaluation setting is standard in the Meta-RL literature (Yu et al., 2017; Humplik et al., 2019) but is not the only way to evaluate agents on unseen tasks in the meta-RL literature. Indeed, several works have considered that given a new task, an agent is given an amount of "free" interactions episodes or steps to perform system identification, then is evaluated on the cumulative reward on one (Bharadhwaj et al., 2019; Rakelly et al., 2019) or several execution episodes (Liu et al., 2020). This is different to what we study here where the agent has to identify the task to solve and solved it within one episode, the reward of the agent being considered during all these steps.

**Online Adaptation with Deep RL.** In the previous section we mentioned that the best strategy corresponds to the optimal policy of the associated POMDP. Since the belief state $b_t$ is a sufficient statistic of the history $\tau_t$, POMDP policies takes the form $\pi(a_t|\tau_t) = \pi(a_t|s_t, b_t)$. While it is impractical to compute the exact belief state even for toy discrete problems, approximations can be learnt using Recurrent Neural Networks (RNNs) (Bakker, 2001; Heess et al., 2015). RNN-based policies are trained to maximize the cumulative reward and do not leverage task descriptors at train time. While this class of policies can represent rich exploratory strategies, their large training complexity makes them impractical.

In order to reduce the training complexity of RNN policies, existing strategies have constrained the set of possible exploratory behaviors by leveraging privileged information about the task. Probe-Then-Exploit (PTE) (e.g. Zhou et al., 2019) works in two phases. First, it executes a pure exploratory policy with the objective of identifying the underlying task $\mu$, i.e. maximizing the likelihood of the task, then runs the optimal policy associated to the estimated task. Both the probing and the informed policies are learned using task descriptors, leading to a much more efficient training process. PTE has two main limitations. First, similarly to explore-then-commit approaches in bandits (e.g. Garivier et al., 2016), the exploration can be suboptimal because it is not reward-driven: valuable time is

wasted to estimate unnecessary information. Second, the switch between probing and exploiting is hard to tune and problem-dependent.

Thompson Sampling (TS) (Thompson, 1933) leverages randomization to mix exploration and exploitation. Similarly to the belief state of an RNN policy, TS maintains a distribution over task descriptors that represents the uncertainty on the current task given $\tau_t$. The policy samples a task from the posterior and executes the corresponding informed policy for several steps. Training is limited to learning informed policies together with a maximum likelihood estimator to map trajectories to distributions over tasks. This strategy proved successful in a variety of problems (e.g. Chapelle & Li, 2011; Osband & Roy, 2017). However, as shown in Fig. 1, TS cannot represent certain probing policies because it is constrained to executing informed policies. Another drawback of TS approaches is that the re-sampling frequency needs to be carefully tuned.

The Task Inference (TI) approach (Humplik et al., 2019) is a RNN trained to simultaneously learn a good policy and predict the task descriptor $\mu$. Denoting by $m : H \rightarrow Z$ the mapping from histories to a latent representation of the belief state ($Z \subseteq \mathbb{R}^d$), the policy $\pi(a_t|z_t)$ selects the action based on the representation $z_t = m(\tau_t)$ constructed by the RNN. During training, $z_t$ is also used to predict the task descriptor $\mu$, using the *task-identification* module $g : Z \rightarrow \mathcal{M}$. The overall objective is:

$$\mathbb{E}\Big[\sum_{t=1}^{|\tau|} \gamma^{t-1} r_t^\mu \Big| \pi\Big] + \beta \mathbb{E}\Big[\sum_{t=1}^{|\tau|} \ell(\mu, g(z_t)) \Big| \pi\Big] \tag{2}$$

where $\ell(\mu, g(z_t))$ is the log-likelihood of $\mu$ under distribution $g(z_t)$. The auxiliary loss is meant to structure the memory of the RNN $m$ rather than be an additional reward for the policy, so training is done by ignoring the effect of $m$ on $\pi$ when computing the gradient of the auxiliary loss with respect to $m$. Humplik et al. (2019) proposed two variants, AuxTask and TI, described in Fig. 2 (b) and (c). In TI, the gradient of the policy sub-network is not backpropagated through the RNN (the dashed green arrow in Fig. 2c, and the policy subnetwork receives the original state features as additional input. For both AuxTask and TI, the training of $\pi$ in TI is purely reward-driven, so they do not suffer from the suboptimality of PTE/TS. However, in contrast to PTE/TS, they do not leverage the smaller sample complexity of training informed policies, and the auxiliary loss is defined over the whole value of $\mu$ while only some dimensions may be relevant to solve the task.

**Learning Task Embeddings**   While in principle the minimal requirement for the approaches above is access to *task identifiers*, i.e. one-hot encodings of the task, these approaches are sensitive to the encoding on task descriptions, and prior knowledge on them. In particular, irrelevant variables have a significant impact on PTE approaches since the probing policy aims at identifying the task. For instance, an agent might waste time reconstructing the full $\mu$ when only part of $\mu$ is needed to act optimally w.r.t the reward. Moreover, TS, TI and AuxTask are guided by a prior distribution over $\mu$ that has to be chosen by hand to fit the ground-truth distribution of tasks. Rakelly et al. (2019) proposed to use a factored Gaussian distribution over transitions as a task embedding architecture rather than a RNN.

Several approaches have been proposed to learn task embeddings (Gupta et al., 2018; Rakelly et al., 2019; Zintgraf et al., 2019; Hausman et al., 2018). The usual approach is to train embeddings of task identifiers jointly with the policies. Humplik et al. (2019) mentions using TI with task embeddings, but the embeddings are pre-trained separately, which requires either additional interactions with the environment or expert traces. Nonetheless, we show in our experiments that TI can be used with task descriptors, considering task prediction as a multiclass classification problem.

**Summary of the contributions**   As for RNN/TI, IMPORT learns an RNN policy to maximize cumulative reward, with no decoupling between probing and exploitation. As such, our approach does not suffer from scheduling difficulties instrinsic to PTE/TS approaches. On the other hand, similarly to PTE/TS and contrarily to RNN/TI, IMPORT leverages the fast training of informed policies through a joint training of an RNN and an informed policy. In addition, IMPORT does not rely on probabilistic models of task descriptors. Learning task embeddings makes the approach robust to irrelevant task descriptors contrary to TI, makes IMPORT applicable when only task identifiers are available and able to better generalize when few training tasks are available.'

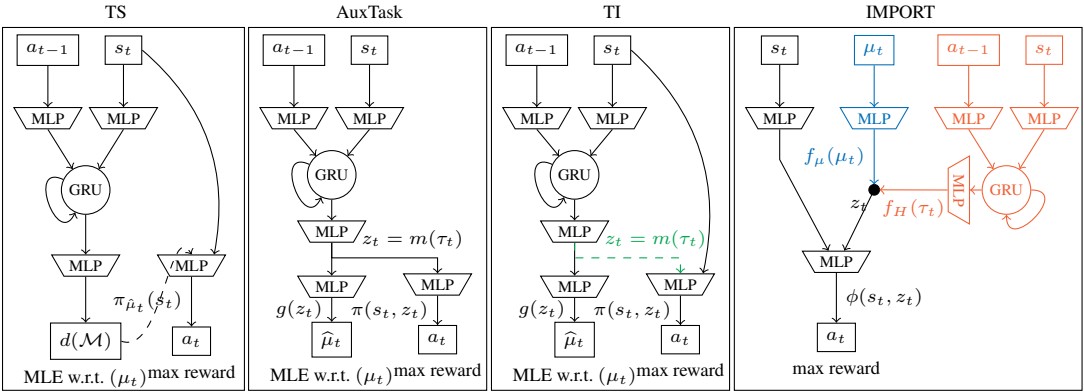

Figure 2: Representation of the different architectures. IMPORT is composed of two models sharing parameters: The (black+blue) architecture is the informed policy $\pi_\mu$ optimized through **(B)** while the (black+red) architecture is the history-based policy $\pi_H$ (used at test time) trained through **(A)+(C)**.

---

**Algorithm 1** IMPORT Training

Initialize $\sigma, \omega, \theta$ randomly
**for** $k = 1, \ldots, K$ **do**
    **if** k is odd **then**
        Collect $M$ transitions following $\pi_H$
        Update $\sigma, \omega$ and the parameters of the value function of **(A)** based on objective **(A)** + **(C)**
    **else**
        Collect $M$ transitions following $\pi_\mu$
        Update $\sigma, \theta, \omega$ and the parameters of the value function of **(B)** based on objective **(B)** + **(C)**
    **end if**
**end for**

---

## 4 METHOD

In this section, we describe the main components of the IMPORT model (described in Fig. 2), as well as the online optimization procedure and an additional auxiliary loss to further speed-up learning.

Our approach leverages the knowledge of the task descriptor $\mu$ and informed policies to construct a latent representation of the task that is *purely reward driven*. Since $\mu$ is unknown at testing time, we use this informed representation to train a predictor based on a recurrent neural network. To leverage the efficiency of informed policies even in this phase, we propose an architecture *sharing parameters* between the informed policy and the final policy such that the final policy will benefit from parameters learned with privileged information. The idea is to constrain the final policy to stay close to the informed policy while allowing it to perform probing actions when needed to effectively reduce the uncertainty about the task. We call this approach InforMed POlicy RegularizaTion (IMPORT).

Formally, we denote by $\pi_\mu(a_t|s_t, \mu)$ and $\pi_H(a_t|\tau_t)$ the informed policy and the history-dependent (RNN) policy that is used at test time. The informed policy $\pi_\mu = \phi \circ f_\mu$ is the functional composition of $f_\mu$ and $\phi$, where $f_\mu : \mathcal{M} \to Z$ projects $\mu$ in a latent space $Z \subseteq \mathbb{R}^k$ and $\phi : \mathcal{S} \times Z \to \mathcal{A}$ selects the action based on the latent representation. The idea is that $f_\mu(\mu)$ captures the relevant information contained in $\mu$ while ignoring dimensions that are not relevant for learning the optimal policy. This behavior is obtained by training $\pi_\mu$ directly to maximize the task reward $r^\mu$.

While $\pi_\mu$ leverages the knowledge of $\mu$ at training time, $\pi_H$ acts based on the sole history. To encourage $\pi_H$ to behave like the informed policy while preserving the ability to probe, $\pi_H$ and $\pi_\mu$ share $\phi$, the mapping from latent representations to actions. We thus define as $\pi_H = \phi \circ f_H$ where $f_H : \mathcal{H} \to Z$ encodes the history into the latent space. By sharing the policy head $\phi$, the approximate belief state constructed by the RNN is mapped to the same latent space as $\mu$. When the uncertainty about the task is small, $\pi_H$ then benefits from the joint training with $\pi_\mu$.

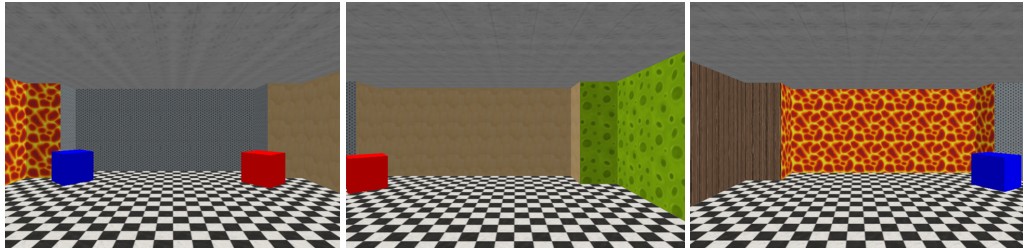

Figure 3: Maze 3D. The goal is either located at the blue or the red box. When the back wall (i.e. not observed in the leftmost image) has a wooden texture, the correct goal is the blue box, whereas if the texture is green, the red box is the goal.

More precisely, let $\theta, \omega, \sigma$ the parameters of $\phi, f_H$ and $f_\mu$ respectively, so that $\pi_\mu^{\sigma\theta}(a_t|s_t, \mu) = \phi^\theta \circ f_\mu^\sigma = \phi^\theta(a_t|s_t, f_\mu^\sigma(\mu))$ and $\pi_H^{\omega\theta}(a_t|\tau_t) = \phi^\theta \circ f_H^\omega = \phi^\theta(a_t|s_t, f_H^\omega(\tau_t))$. The goal of IMPORT is to maximize over $\theta, \omega, \sigma$ the objective function defined in Eq. 3.

$$
\underbrace{\mathbb{E}\left[\sum_{t=1}^{|\tau|} \gamma^{t-1} r_t^\mu \,\middle|\, \pi_H^{\omega,\theta}\right]}_{\textbf{(A)}} + \underbrace{\mathbb{E}\left[\sum_{t=1}^{|\tau|} \gamma^{t-1} r_t^\mu \,\middle|\, \pi_\mu^{\sigma,\theta}\right]}_{\textbf{(B)}} \quad \underbrace{-\beta\mathbb{E}\left[\sum_{t=1}^{|\tau|} D\Big(f_\mu(\mu), f_H(\tau_t)\Big)\right]}_{\textbf{(C)}} \tag{3}
$$

**Speeding Up the Learning.** The optimization of **(B)** in Eq. 3 produces a reward-driven latent representation of the task through $f_\mu$. In order to encourage the history-based policy to predict a task embedding close to the one predicted by the informed policy, we augment the objective with an auxiliary loss **(C)** weighted by $\beta > 0$. $D$ is the squared 2-norm in our experiments. Note that because we treat the objective **(C)** as an auxiliary loss, only the average gradient of $D$ with respect to $f_H$ is backpropagated, ignoring the effect of $f_H$ on $\pi_H$. The expectation of **(C)** is optimized over trajectories generated using $\pi_H^{\omega,\theta}$ and $\pi_\mu^{\sigma,\theta}$, respectively used to compute **(A)** and **(B)**.

**Optimization.** IMPORT is trained using Advantage Actor Critic (A2C) (Mnih et al., 2016) with generalized advantage estimation (GAE) (Schulman et al., 2015). There are two value functions[1], one for each objective **(A)** and **(B)**. The algorithm is summarized in Alg. 1. Each iteration collects a batch of $M$ transitions using either $\pi_H$ or $\pi_\mu$.[2] If the batch is sampled according to $\pi_H$, we update with A2C-GAE the parameters of the policy $\omega$ and $\theta$ according to both objectives **(A)** and **(C)**, as well as the parameters of the value function associated to objective **(A)**. If the batch is sampled according to $\pi_\mu$, we update with A2C-GAE the parameters of the policy $\sigma$ and $\theta$ according to both objectives **(B)** and **(C)**, as well as the parameters of the value function associated to objective **(B)**.

## 5 EXPERIMENTS

We performed experiments on five environments. The **CartPole** and **Acrobot** environments from OpenAI Gym (Brockman et al., 2016), where the task descriptor $\mu$ represents parameters of the physical system, e.g., the weight of the cart, the size of the pole, etc. The dimension of $\mu$ is 5 for Cartpole and 7 for Acrobot. The entries of $\mu$ are normalized in $[-1, 1]$ and sampled uniformly. These environments provide basic comparison points where the optimal exploration/exploitation policy is relatively straightforward, since the dynamics can be inferred from a few actions. The **Bandit** environment is a standard Bernoulli multi-armed bandit problem with $K$ arms. The vector $\mu \in \mathbb{R}^K$ denotes the probability of success of the independent Bernoulli distributions. Each dimension of $\mu$ is sampled uniformly between 0 and 0.5, the best arm is randomly selected and associated to a probability of 0.9. An episode is 100 arm pulls. At every timestep the agent is allowed to pull an arm

---

[1]In our implementation, the value network is shared and takes as an input either $f\mu(\mu)$ or $f_H(\tau_t)$.

[2]In practice, data collection is multithreaded. We collect 20 transitions per thread with 24 to 64 threads depending on the environment, based on available GPU memory

|  | $N = 10$ | $N = 20$ | $N = 100$ | $N = 10$ | $N = 20$ | $N = 100$ |
|---|---|---|---|---|---|---|
| RNN | 73.4(4.8) | 92.9(1.3) | 87.5(0.2) | | | |
| | Using $\mu$ at train time | | | Using task identifier at train time | | |
| IMPORT | **94.4(0.7)** | **94.8(0.8)** | **95.3(0.4)** | **92.8(0.8)** | **95.5(0.4)** | 95.1(1.0) |
| AuxTask | 91.0(0.7) | 92.0(1.9) | 92.6(0.7) | 90.5(1.8) | 91.2(1.7) | 94.3(0.7) |
| TI | 91.5(0.6) | 94.4(0.4) | 94.6(0.3) | 90.8(0.5) | 90.7(1.2) | **97.0(0.2)** |
| TS | 88.7(4.2) | 87.3(2.5) | 91.3(1.7) | 85.9(2.4) | 90.1(1.3) | 91.0(0.5) |

Table 1: CartPole with different number $N$ of training tasks. Note that RNN does not $\mu$ at train time.

|  | $S = 1$ | $S = 3$ | $S = 5$ | $S = 1$ | $S = 3$ | $S = 5$ |
|---|---|---|---|---|---|---|
| Size of $\mu$: | 5 | 60 | 150 | 5 | 60 | 150 |
| RNN | 74.0(5.8) | 77.0(1.1) | 66.9(0.7) | | | |
| | Using $\mu$ | | | Using task identifier | | |
| IMPORT | **89.2(0.1)** | **79.4(0.6)** | **72.7(0.1)** | **89.1(0.1)** | **79.7(0.1)** | **74.0(0.3)** |
| AuxTask | 84.7(1.1) | 76.2(2.0) | 66.9(1.2) | 87.7(0.8) | 78.0(0.4) | 72.4(0.5) |
| TI | 78.7(0.9) | 76.5(0.3) | 68.3(0.8) | 85.3(0.8) | 77.5(1.1) | 70.9(0.2) |
| TS | 85.9(0.5) | 64.1(0.6) | 60.7(0.1) | 84.1(2.5) | 65.7(0.3) | 60.2(1.0) |

Table 2: Result over Tabular-MDP with $S$ states and $A = 5$ actions, trained over $N = 100$ tasks.

in [1, K] and observes the resulting reward. Although relatively simple, this environment assesses the ability of algorithms to learn nontrivial probing/exploitation strategies. The **Tabular MDP** environment is a finite MDP with $S$ states and $A$ actions such that the transition matrix is sampled from a flat Dirichlet distribution, and the reward function is sampled from a uniform distribution in $[0, 1]$ as in Duan et al. (2016). In that case, $\mu$ is the concatenation of the transition and the reward functions, resulting in a vector of size $S^2 A + SA$. This environment is much more challenging as $\mu$ is high-dimensional, there is nearly complete uncertainty on the task at hand and each task is a reinforcement learning problem. Finally, the **Maze 3D** environment is a 3D version of the toy problem depicted in Fig. 1, implemented using gym-miniworld (Chevalier-Boisvert, 2018). It has three discrete actions (*forward, left, right*) and the objective is to reach one of the two possible goals (see Figure 15 in appendix), resulting in a reward of $+1$ (resp. $-1$) when the correct (resp. wrong) goal is reached. The episode terminates when the agent touches a box or after 100 steps. The agent always starts at a random position, with a random orientation. The information about which goal to reach at each episode is encoded by the use of two different textures on the wall located at the opposite side of the maze w.r.t. the goals. This domain allows to evaluate the models when observations are high dimensional ($3 \times 60 \times 60$ RGB images). The maximum episode length is 100 on CartPole, Bandit, Tabular-MDP and Maze3D, and 500 on Acrobot. To evaluate the ability of IMPORT and the baselines to deal with different types of task descriptors $\mu$, we also perform experiments on CartPole and Tabular-MDP in the setting where $\mu$ is only a task identifier (i.e., a one-hot vector representing the index of the training task) which is a very weak supervision available at train time.

We compare to previously discussed baselines. First, a vanilla RNN policy (Heess et al., 2015) using GRUs that never uses $\mu$. Second, we compare to TS, TI and AuxTask, with $\mu$ only observed at train time, similarly to IMPORT. For TS, at train time, the policy conditions on the true $\mu$, whereas at test time, the policy conditions on an estimated $\widehat{\mu}$ resampled from the posterior every $k$ steps where $k \in \{1, 5, 10, 20\}$. On bandits, UCB (Auer, 2002) with tuned exploration parameters is our topline.

**Implementation details** Contrarily to IMPORT, TS, TI and AuxTask are based on maximizing the log-likelihood of $\mu$. When using informative task descriptors (i.e. a vector of real values), the log-likelihood uses a Gaussian distribution with learnt mean and diagonal covariance matrix. For the bandit setting, we have also performed experiments using a beta distribution which may be more relevant for this type of problem. When using task identifiers, a multinomial distribution is used. All approaches are trained using A2C with Generalized Advantage Estimation (Mnih et al., 2016;

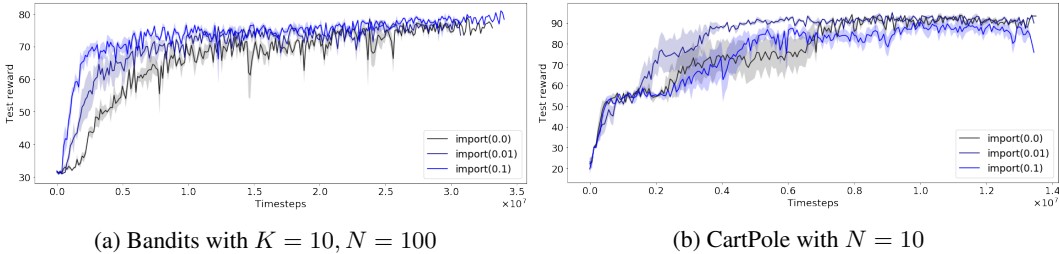

(a) Bandits with $K = 10, N = 100$       (b) CartPole with $N = 10$

Figure 4: Test performance of IMPORT for different values of $\beta$ from Eq. 3

Schulman et al., 2015). The precise values of the hyper-parameters and architectures are given in Appendix B.2.

All approaches use similar network architectures with the same number of hidden layers and units.

**Evaluation** The meta-learning scenario is implemented by sampling $N$ training tasks, $N$ validation tasks and $10,000$ test tasks with no overlap between task sets (except in Maze3D where there is only two possible tasks). Each sampled training task is given a unique identifier. Each model is trained on the training tasks, and the best model is selected on the validation tasks. We report the performance on the test tasks, averaged over three trials with different random seeds, corresponding to different sets of train/validation/test tasks. Training uses a discount factor, but for validation and test, we compute the undiscounted cumulative reward on the validation/test tasks. The learning curves show test reward as a function of the environment steps. They are the average of the three curves associated to the best validation model of each of the three seeds used to generate different tasks sets.

**Overall performances.** IMPORT performs better than its competitors in almost all the settings. For instance, on CartPole with 10 tasks (see Table 1), our model reaches $94.4$ reward while TI reaches only $91.5$. Qualitatively similar results are found on Acrobot (Table 5 in Appendix), as well as on Bandit with 20 arms (Table 3), even though AuxTask performs best with only 10 arms. IMPORT particularly shines when $\mu$ encodes complex information, as on Tabular-MDP (see Table 2) where it outperforms all baselines in all settings. By varying the number of training tasks on CartPole and Acrobot, we also show that IMPORT's advantage over the baselines is larger with fewer training tasks. In all our experiments, as expected, the vanilla RNN performs worse than the other algorithms.

**Sample Efficiency.** Figure 5 shows the convergence curves on CartPole with 10 and 100 training tasks and are representative of what we obtain on other environments (see Appendix). IMPORT tends to converge faster than the baselines. We also observe a positive effect of using the auxiliary loss ($\beta > 0$) on sample efficiency, in particular with few training tasks. Note that using the auxiliary loss is particularly efficient in environments where the final policy tends to behave like the informed on.

**Influence of $\mu$.** The experiments with uninformative $\mu$ (i.e., task identifiers) reported in Table 1 and 2 for CartPole and Tabular-MDP respectively show that the methods are effective even when the task descriptors do not include any prior knowledge. In the two cases, IMPORT can use these tasks descriptors to generalize well. Moreover, experimental results on CartPole (Fig. 11) and Tabular MDP (Fig. 17) suggest that when $\mu$ is a vector of features (and not a task identifier only), it improves sample efficiency but does not change the final performance. This can be explained by the fact that informed policies are faster to learn with features in $\mu$ since, in that case, $\mu$ is capturing similarities between tasks. Equivalent performance of IMPORT on both types of task descriptors is observed and shows that our method can deal with different (rich and weak) task descriptors. We further analyze the impact of the encoding of $\mu$ on the models, by using non-linear projections of the informative $\mu$ to change the shape of the prior knowledge. Figure 5c shows the learning curves of TI and IMPORT on CartPole with task identifiers, the original $\mu$ and polynomial expansions of $\mu$ of order 2 and 3, resulting in 21 and 56 features. IMPORT's task embedding approach is robust to the encoding of $\mu$, while TI's log-likelihood approach underperforms with the polynomial transformation.

**Task embeddings.** To have a qualitative assessment of the task embedding learnt by IMPORT, we consider a bandit problem with 10 arms and embedding dimension 16. Figure 6 shows the clusters of task embeddings obtained with t-SNE (Maaten & Hinton, 2008). Each cluster maps to an optimal arm, showing that IMPORT structures the embedding space based on the relevant information. In

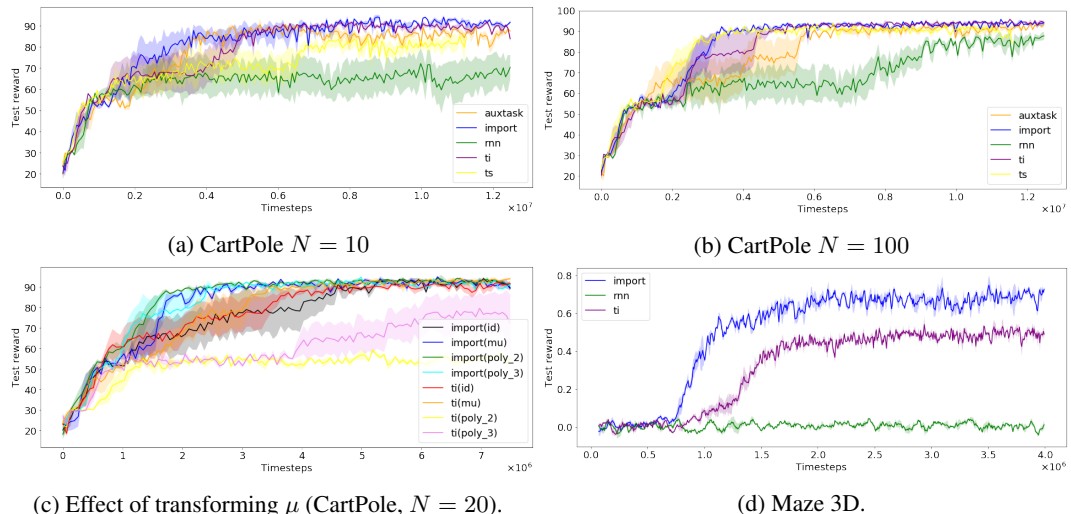

(a) CartPole $N = 10$

(b) CartPole $N = 100$

(c) Effect of transforming $\mu$ (CartPole, $N = 20$).

(d) Maze 3D.

Figure 5: Learning curves on CartPole (a and b) and Maze3D (d) test tasks. Figure (c) studies the impact of the structure of the task descriptor on the performances of TI and IMPORT in CartPole.

|                    | $K = 10$     | $K = 20$       |
|--------------------|--------------|----------------|
| IMPORT             | 77.5(0.2)    | **56.6(0.1)**  |
| AuxTask (Gaussian) | 78.7(0.4)    | 50.5(1.6)      |
| AuxTask (Beta)     | 78.2(0.7)    | 37.1(0.6)      |
| RNN                | 73.6(0.7)    | 32.1(1.2)      |
| TI (Gaussian)      | 73.7(1.6)    | 41.4(2.4)      |
| TI (Beta)          | **79.5(0.1)**| 53.3(2.4)      |
| TS (Gaussian)      | 50.4(0.4)    | 38.8(2.0)      |
| TS (Beta)          | 41.3(1.5)    | 36.3(1.1)      |
| UCB                | 78.5(0.3)    | 68.2(0.4)      |

Table 3: Bandits performance for $K = 10$ and $K = 20$ arms, with $N = 100$ training tasks.

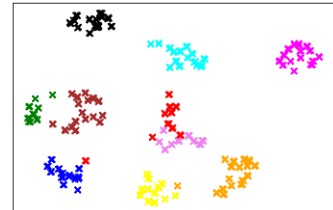

Figure 6: Task embeddings learnt on Bandit (10 arms). Colors indicate the best arm.

addition, we have studied the influence of the $\beta$ hyperparameter from Eq. 3 (in Fig. 4 and Section D). It shows that the auxiliary loss helps to speed-up the learning process, but is not necessary to achieve great performance.

**High dimensional input space.** We show the learning curves on the Maze3D environment in Figure 5d. IMPORT is succeeding in 90% of cases (reward $\approx 0.8$), while TI succeeds only in 70% of cases. This shows that IMPORT is even more effective with high-dimensional observations (here, pixels). IMPORT and TI benefit from knowing $\mu$ at train time, which allows them to rapidly identify that the wall texture behind the agent is informative, while the vanilla RNN struggles and reaches random goals. TS is not reported since this environment is a typical failure case as discussed in Fig.1.

**Additional results.** In Appendix C.1, we show that IMPORT outperforms TI by a larger margin when the task embedding dimension is small. We also show that IMPORT outperforms its competitors in dynamic environments, i.e., when the task changes during the episode.

## 6 CONCLUSION

We proposed a new policy architecture for meta reinforcement learrning. The IMPORT model is trained only on the reward objective, and leverages the informed policy to discover effective trade-offs between exploration and exploitation. It is thus able to learn better strategies than Thompson Sampling approaches, and faster than recurrent neural network policies and Task Inference approaches.

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

## A    THE IMPORT ALGORITHM

The algorithm is described in details in Algorithm 2. In our implementation, the value function network used for **(A)** and **(B)** is the same, i.e. shared. We specialize the input, i.e. for **(A)** the input will be $(s_t, f_H(\tau_t))$ and $(s_t, f_\mu(\mu_t))$ for **(B)**.

---

**Algorithm 2** Details of IMPORT Training

---

Initialize $\sigma, \omega, \theta, \nu$ arbitrarily
Hyperparameters:    Number of iterations    $K$,    Number of transitions per update steps    $M$, discount factor $\gamma$,   GAE parameter   $\gamma_{GAE}$,   Adam learning rate   $\eta$,   weighting of the **(C)** objective $\beta$, weighting of the entropy objective $\lambda_h$, weighting of the critic objective $\lambda_c$
$Optim = Adam(\eta)$
**for** $k = 1, \ldots, K$ **do**
  **if** $k$ is odd **then**
    Collect $M$ transitions according to $\pi_H$ in buffer $B_H$.
  **else**
    Collect $M$ transitions according to $\pi_\mu$ in buffer $B_\mu$.
  **end if**
  $\delta_\sigma, \delta_\omega, \delta_\theta = 0, 0, 0$

  $R^\mu \leftarrow \text{compute\_gae\_returns}(B_\mu, \gamma_{GAE})$
  $R^H \leftarrow \text{compute\_gae\_returns}(B_H, \gamma_{GAE})$

  $\delta_{\theta,\omega} \mathrel{+}= \frac{1}{|B_H|} \sum_{b \in B_H} \sum_{t=1}^T [R_t^{\mu,b} - V_\nu(s_t^b, z_t^b)] \nabla_{\theta,\omega} \log \pi_H(a_t^b | s_t^b, z_t^b)$
  $\delta_{\theta,\omega} \mathrel{+}= \frac{\lambda_h}{|B_H|} \sum_{b \in B_H} \sum_{t=1}^T \nabla_{\theta,\omega} H\big(\pi_H(a_t^b | s_t^b, z_t^b)\big)$
  $\delta_\omega \mathrel{-}= \frac{2\beta}{|B_H|} \sum_{b \in B_H} \sum_{t=1}^T [f_H^\omega(s_t^b, z_t^b) - f_\mu(s_t^b, \mu_t^b)] \nabla_\omega f_H^\omega(s_t^b, z_t^b)$
  $\delta_\nu \mathrel{-}= \frac{2\lambda_c}{|B_H|} \sum_{b \in B_H} \sum_{t=1}^T [R_t^{H,b} - V_\nu(s_t^b, z_t^b)] \nabla_\nu V_\nu(s_t^b, z_t^b)$

  $\delta_{\theta,\sigma} \mathrel{+}= \frac{1}{|B_\mu|} \sum_{b \in B_\mu} \sum_{t=1}^T [R_t^{H,b} - V_\nu(s_t^b, \mu_t^b)] \nabla_{\theta,\sigma} \log \pi_\mu(a_t^b | s_t^b, \mu_t^b)$
  $\delta_{\theta,\sigma} \mathrel{+}= \frac{\lambda_h}{|B_\mu|} \sum_{b \in B_\mu} \sum_{t=1}^T \nabla_{\theta,\sigma} H\big(\pi_\mu(a_t^b | s_t^b, \mu_t^b)\big)$
  $\delta_\nu \mathrel{-}= \frac{2\lambda_c}{|B_\mu|} \sum_{b \in B_\mu} \sum_{t=1}^T [R_t^{\mu,b} - V_\nu(s_t^b, \mu_t^b)] \nabla_\nu V_\nu(s_t^b, \mu_t^b)$

  $\theta \leftarrow Optim(\theta, \delta_\theta)$
  $\omega \leftarrow Optim(\omega, \delta_\omega)$
  $\sigma \leftarrow Optim(\sigma, \delta_\sigma)$
  $\nu \leftarrow Optim(\nu, \delta_\nu)$
**end for**

---

## B    IMPLEMENTATION DETAILS

### B.1    DATA COLLECTION AND OPTIMIZATION

We focus on on-policy training for which we use the actor-critic method A2C (Mnih et al., 2016) algorithm with generalized advantage estimation. We use a distributed execution to accelerate experience collection. Several worker processes independently collect trajectories. As workers progress, a shared replay buffer is filled with trajectories and an optimization step happens when the buffer's capacity $bs$ is reached. After model updates, replay buffer is emptied and the parameters of all workers are updated to guarantee synchronisation.

### B.2    NETWORK ARCHITECTURES

The architecture of the different methods remains the same in all our experiments, except that the number of hidden units changes across considered environments and we consider convolutional neural networks for the Maze3d environment. A description of the architectures of each method is given in Fig. 2.
Unless otherwise specified, MLP blocks represent single linear layers activated with a $tanh$ function and their output size is $hs$. All methods aggregate the trajectory into an embedding $z_t$ using a GRU with hidden size $hs$. Its input is the concatenation of representations of the last action $a_{t-1}$ and

| HPs | CartPole | Acrobot | Bandits | TMDP | Maze3d |
|---|---|---|---|---|---|
| $E$ | 16 | 128 | 16 | 16 | 32 |
| $Tr$ | 20 | 20 | 20 | 20 | 20 |
| $hs$ | 16 | 32 | 16 | 64 | 32 |
| $hs_\mu$ | $\{2,4,8,16\}$ | $\{2,4,8,16\}$ | 16 | $\{16,32\}$ | $\{2,16,32\}$ |
| $\gamma$ | 0.95 | 0.95 | 0.90 | 0.90 | 0.90 |
| $\lambda_h$ | $\{1.,1e^{-1}\}$ | | | | $\{1e^{-1},1e^{-2},1e^{-3}\}$ |
| $\gamma_{GAE}$ | $\{0.0,1.0\}$ | | | | |
| clip gradient | 40 | | | | |
| $\eta$ | $\{1e^{-3},3e^{-4}\}$ | | | | |
| $\lambda_c$ | $\{1.,1e^{-1},1e^{-2}\}$ | | | | |
| $\beta$ | $\{1e^{-1},1e^{-2},0.\}$ | | | | |

Table 4: Hyperparameters tested per environments. At each training epoch, we run our agent on $E$ environments in parallel collecting $Tr$ transitions on each of them resulting in batches of $M = E * Tr$ transitions.

current state $s_t$ obtained separately. Actions are encoded as one-hot vectors. When episodes begin, we initialize the last action with a vector of zeros. For bandits environments, the current state corresponds to the previous reward. TS uses the same GRU architecture to aggregate the history into $z_t$.

All methods use a $softmax$ activation to obtain a probability distribution over actions.
The use of the hidden-state $z_t$ differs across methods. While **RNNs** only use $z_t$ as an input to the policy and critic, both **TS** and **TI** map $z_t$ to a belief distribution that is problem-specific, e.g. Gaussian for control problems, Beta distribution for bandits, and a multinomial distribution for Maze and CartPole-task environments. For instance, $z_t$ is mapped to a Gaussian distribution by using two MLPs whose outputs of size $|\mu|$ correspond to the mean and variance. The variance values are mapped to $[0, 1]$ using a $sigmoid$ activation.

**IMPORT** maps $z_t$ to an embedding $f_H$, whereas the task embedding $f_\mu$ is obtained by using a $tanh$-activated linear mapping of $\mu_t$. Both embeddings have size $hs_\mu$, tuned by cross-validation onto a set of validation tasks. The input of the shared policy head $\phi$ is the embedding associated with the policy to use, i.e. either $f_H$ when using $\pi_H$ or $f_\mu$ when using $f_\mu$.

For the Maze3d experiment and in all methods, we pre-process the pixel input $s_t$ with three convolutional layers (with output channels 32, stride is 2 and respective kernel sizes are 5, 5 and 4) and LeakyReLU activation. We also use a batch-norm after each convolutional layer. The output is flattened, linearly mapped to a vector of size $hs$ and $tanh$-activated.

## C  EXPERIMENTS

In this section, we explain in deeper details the environments and the set of hyper-parameters we considered. We add learning curves of all experiments to supplement results from Table 1, 2, 3 and 5 in order to study sample efficiency.

**Task descriptor.**  Note that for CartPole and Acrobot $\mu$ is normalized to be in $[-1, 1]^D$ where $D$ is the task descriptor dimension. The task distribution $q$ is always uniform, see the description of the environments for details. For experiments with task identifiers, we associate to each sampled task an integer value corresponding to the order of generation, and encode it usong a one-hot vector.

**Hyperparameters.**  Hyperparameter ranges are specified in Table 4. For TS, we consider sampling $\mu$ from the posterior dynamics distribution every $k$ steps with $k \in \{1, 5, 10, 20\}$.

## C.1 CARTPOLE.

We consider the classic CartPole control environment where the environment dynamics change within a set $\mathcal{M}$ ($|\mu| = 5$) described by the following physical variables: gravity, cart mass, pole mass, pole length, magnetic force. Their respective pre-normalized domains are $[4.8, 14.8], [0.5, 1.5], [0.01, 0.19], [0.2, 0.8]$, and $[-10, 10]$. The value of $\mu$ are uniformly sampled. Knowing some components of $\mu$ might not be required to behave optimally. The discrete action space is $\{-1, 1\}$.

Episode length is $T = 100$.

**Final performance and sample efficiency.** Table 1 shows IMPORT's performance is marginally superior to other methods in most settings. Learning curves in Figure 7 allow analyzing the sample efficiency of the different methods. Overall, IMPORT is more sample efficient than other methods in the privileged information $\mu$ setting. Moreover, the use of the auxiliary loss ($\beta > 0$) usually speed-up the learning convergence by enforcing the RNN to quickly produce a coherent embedding. We can see that only sharing parameters ($\beta = 0$) already helps improving over RNNs.

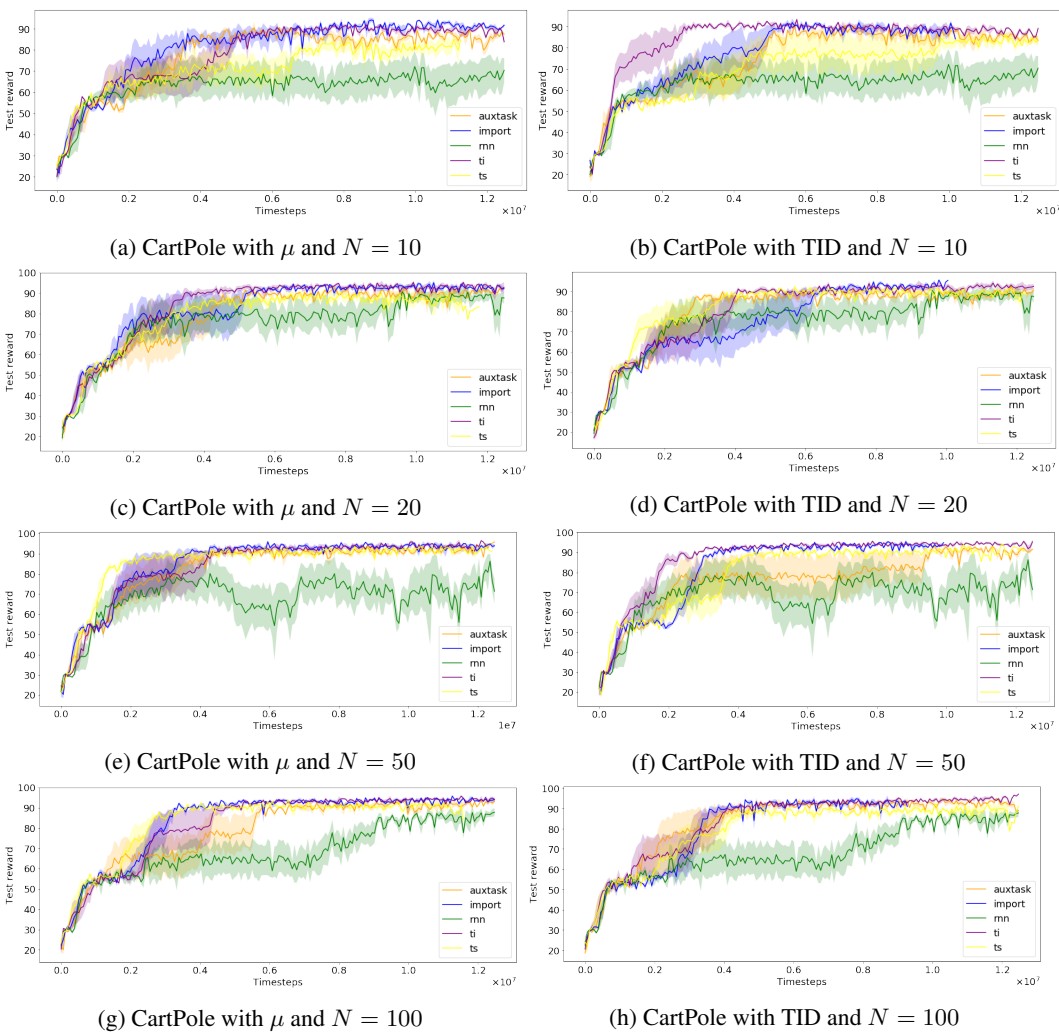

(a) CartPole with $\mu$ and $N = 10$

(b) CartPole with TID and $N = 10$

(c) CartPole with $\mu$ and $N = 20$

(d) CartPole with TID and $N = 20$

(e) CartPole with $\mu$ and $N = 50$

(f) CartPole with TID and $N = 50$

(g) CartPole with $\mu$ and $N = 100$

(h) CartPole with TID and $N = 100$

Figure 7: Evaluation on CartPole

**Non-stationary environments.** We consider the non-stationary version of CarPole environment where at each timestep, there is a probability $\rho = 0.05$ to sample a new dynamic $\mu$. Table 8 shows that the performance of IMPORT, AuxTask and TI are comparable in these settings.

| Method | $N = 10$ | $N = 100$ |
|--------|----------|-----------|
| AuxTask | 86.4(1.0) | 93.0(0.3) |
| IMPORT | **91.7(0.5)** | 92.7(0.8) |
| RNN | 65.5(4.3) | 89.5(0.6) |
| TI | 88.2(3.9) | **95.5(0.8)** |
| TS | 86.7(1.6) | 92.2(0.7) |

Figure 8: CartPole (non-stationary).

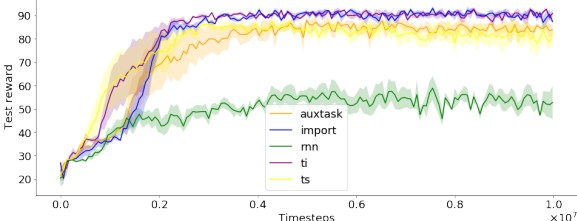

Figure 9: Non-stationary CartPole with $N = 10$

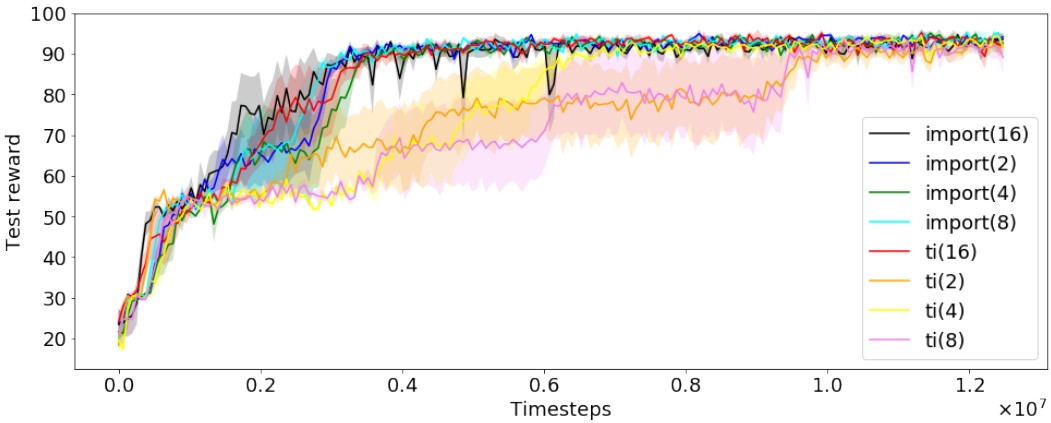

Figure 10: IMPORT and TI with different task embedding representation size on CartPole with $N = 20$

**Size of built embeddings.** We now study the impact of the task embedding representation size. As can be seen from Figure 10, IMPORT's performance remains stable for different representation sizes in $\{2, 4, 8, 16\}$ whereas TI's sample efficiency decreases with this dimension.

**Trajectory and task embeddings.** In Figure 11, we plot both the evolution of $f_H(\tau_t)$ during an episode of the final model obtained training IMPORT with two-dimensional task embeddings on CartPole with **task identifiers** (left) and task embedding $f_\mu(\mu)$ learnt by the informed policy (right). As expected, the history embedding gets close to the task embedding after just a few timesteps (left). Interestingly, task embeddings $f_\mu(\mu)$ are able to capture relevant information from the task. For instance, they are highly correlated with the *magnetic force* which is a very strong factor to "understand" from each new environment to control the system correctly. At the opposite, *gravity* is less correlated since it does not influence the optimal policy – whatever the gravity is, if the pole is on the left, then you have to go right and vice-versa.

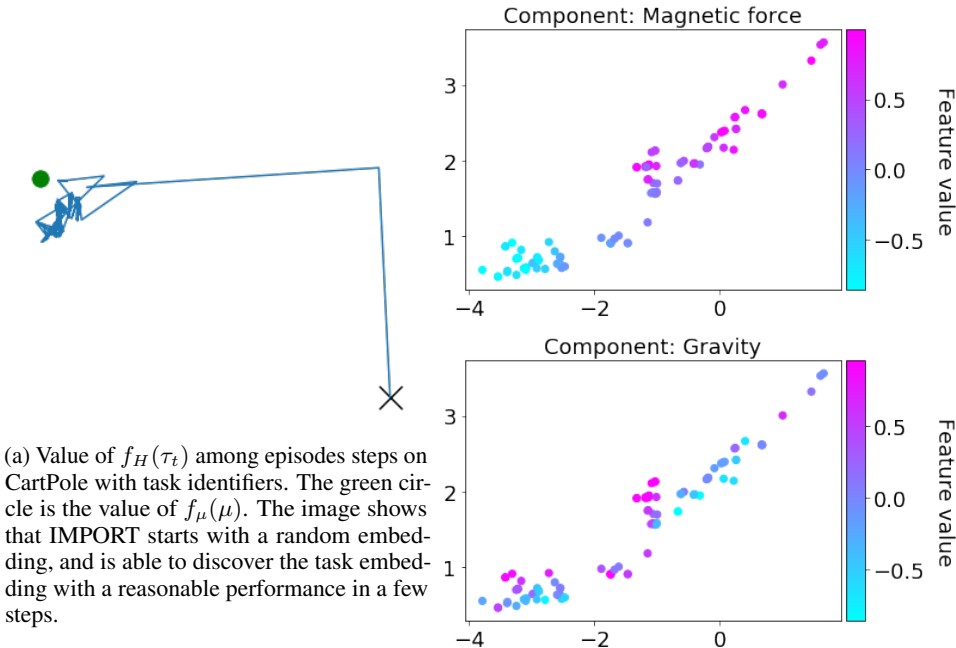

(a) Value of $f_H(\tau_t)$ among episodes steps on CartPole with task identifiers. The green circle is the value of $f_\mu(\mu)$. The image shows that IMPORT starts with a random embedding, and is able to discover the task embedding with a reasonable performance in a few steps.

(b) Task embeddings $f_\mu(\mu)$ for Cartpole with task identifiers. The color of the point corresponds to the value of one of the 'real' physics component of the environment (unknown to the model).

Figure 11: Visualization of task embeddings upon Cartpole

## C.2    ACROBOT

|  | $N = 10$ | $N = 20$ | $N = 50$ | $N = 100$ |
|---|---|---|---|---|
| AuxTask | $-189.0(54.8)$ | $-98.3(1.8)$ | $-103.0(8.0)$ | $-93.6(1.3)$ |
| IMPORT | $\mathbf{-87.2(0.9)}$ | $\mathbf{-92.5(1.3)}$ | $-88.9(1.1)$ | $-88.9(1.6)$ |
| RNN | $-483.6(1.6)$ | $-482.7(4.0)$ | $-480.7(3.5)$ | $-485.0(3.7)$ |
| TI | $-89.7(1.2)$ | $-94.6(0.7)$ | $\mathbf{-87.8(0.8)}$ | $\mathbf{-87.3(1.2)}$ |
| TS | $-101.4(2.0)$ | $-102.1(6.0)$ | $-102.4(2.0)$ | $-102.3(0.8)$ |

Table 5: Acrobot

Acrobot consists of two joints and two links, where the joint between the two links is actuated. Initially, the links are hanging downwards, and the goal is to swing the end of the lower link up to a given height. Environment dynamics are determined by the length of the two links, their masses, their maximum velocity. Their respective pre-normalized domains are $[0.5, 1.5], [0.5, 1.5], [0.5, 1.5], [0.5, 1.5], [3\pi, 5\pi]$ and $[7\pi, 11\pi]$. Unlike CartPole, the environment is stochastic because the simulator applies noise to the applied force. The action space is $\{-1, 0, 1\}$. We also add an extra dynamics parameter which controls whether the action order is inverted, i.e. $\{1, 0, -1\}$, thus $|\mu| = 7$.

Episode length is $500$.

IMPORT outperforms all baselines in settings with small training task sets (Figure 12 and Table 5) and perform similarly to TI on larger training task sets.

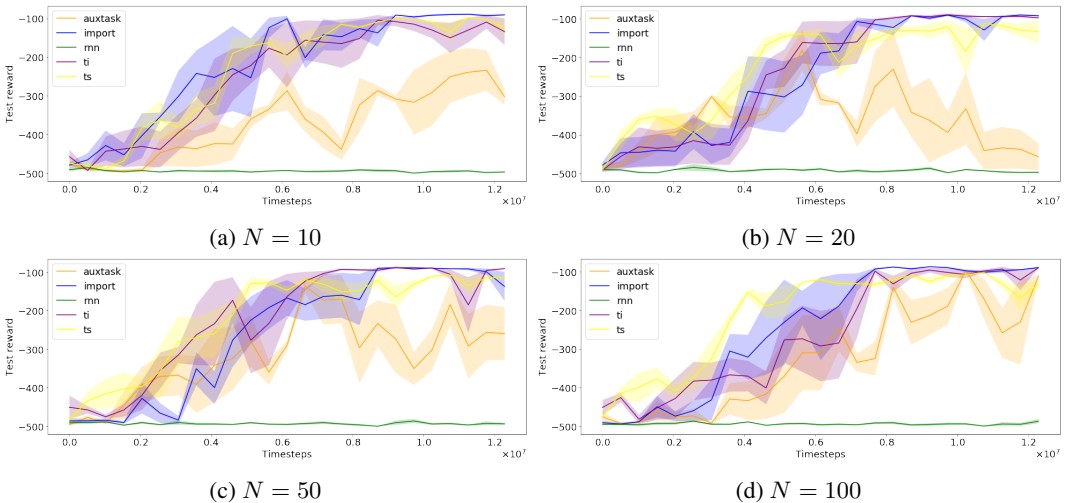

(a) $N = 10$    (b) $N = 20$

(c) $N = 50$    (d) $N = 100$

Figure 12: Performance on Acrobot

## C.3 BANDITS

The **Bandit** environment is a standard Bernoulli multi-armed bandit problem with $K$ arms. The vector $\mu \in \mathbb{R}^K$ denotes the probability of success of the independent Bernoulli distributions. Each dimension of $\mu$ is sampled uniformly between $0$ and $0.5$, the best arm is randomly selected and associated to a probability of $0.9$. Although relatively simple, this environment assesses the ability of algorithms to learn nontrivial exploration/exploitation strategies.

Note that it is not surprising that UCB outperforms the other algorithms in this setting. UCB is an optimal algorithm for MAB and we have optimized it for achieving the best empirical performance. Moreover, IMPORT cannot leverage correlations between tasks since, due to the generation process, tasks are independent.

We visualize the task embeddings learnt by the informed policy in 13.

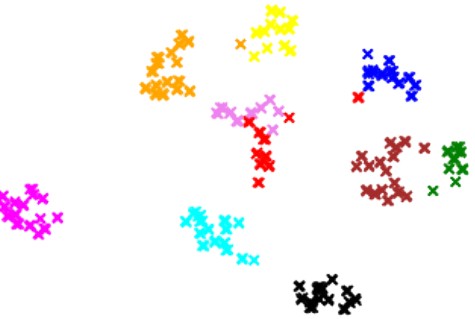

Figure 13: t-SNE of the task embeddings on the bandit problem with $K = 10$.

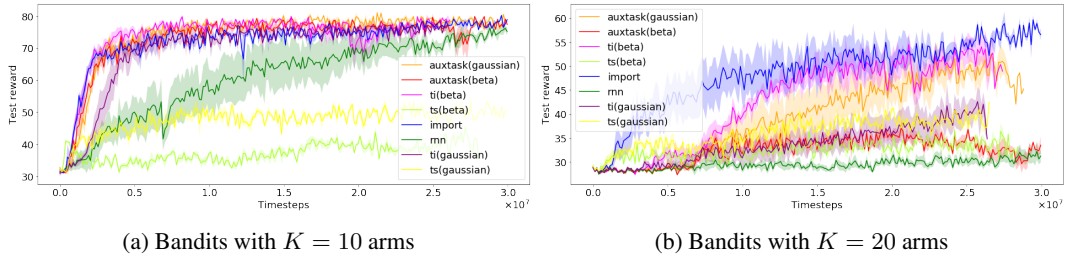

(a) Bandits with $K = 10$ arms  (b) Bandits with $K = 20$ arms

Figure 14: Learning curves on the bandit problem.

## C.4 MAZE3D ENVIRONMENT

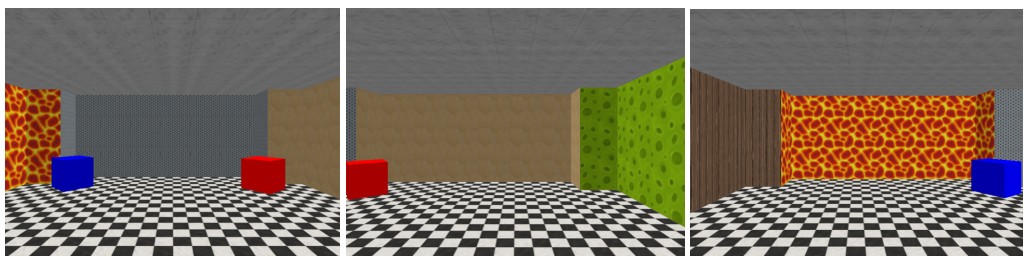

Figure 15: Maze 3D. The goal is either located at the blue or the red box. When the wall on the opposite side of the boxes (i.e. not observed in the leftmost image) has a wooden texture, the correct goal is the blue box, whereas if the texture is green, the red box is the goal.

The **Maze 3D** environment (Figure 15) is a continuous maze problem implemented using gym-miniworld (Chevalier-Boisvert, 2018), with 3 discrete actions (forward, left, right) where the objective is to reach one of the two possible goals, resulting in a reward of $+1$ (resp. $-1$) when the correct (resp. wrong) goal is reached. If a box is touched, the episode ends. The maze's axis range from -40 to 40, the two turn actions (*left*, *right*) modify the angle by 45 degrees, and the *forward action* is a 5 length move. The agent starts in a random position with a random orientation. The information about which goal to reach at each episode is encoded by the use of two different textures on the wall located on the opposite side of the boxes. In this way, the agent cannot simultaneously observe both boxes and the "informative" wall.

This environment allows to evaluate the models in a setting where the observation is a high dimensional space (3x60x60 RGB image). The mapping between the RGB image and the task target in $\{-1, 1\}$ is challenging and the informed policy should provide better auxiliary task targets than TI thanks to the "easy" training of the informed policy.

IMPORT outperforms TI on this environment (Figure 16) in both final performance and sample efficiency.

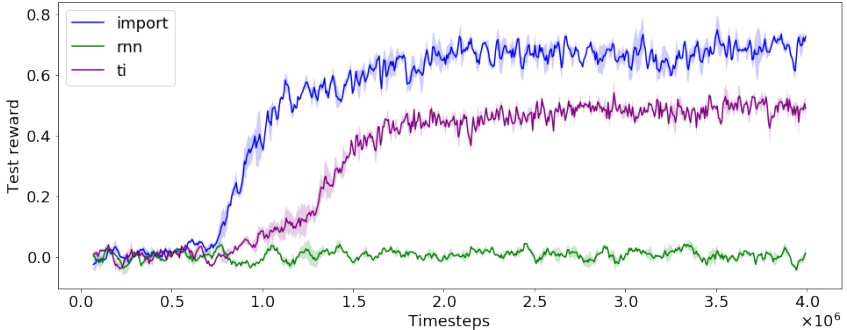

Figure 16: Learning curves on the Maze 3D environment

## C.5 TABULAR MDPs

Tabular MDP (Duan et al., 2016) is a MDP with $S$ discrete states and $A$ actions such that the transition matrix is sampled from a flat Dirichlet distribution, and the reward function is sampled from a uniform distribution in $[0, 1]$. The task identifier $\mu$ is a concatenation of the transition and reward functions resulting in a vector of size $S^2 A + SA$, allowing to test the models with high-dimensional $\mu$.

IMPORT outperforms all baselines in all settings (Figure 17 and Table 2).

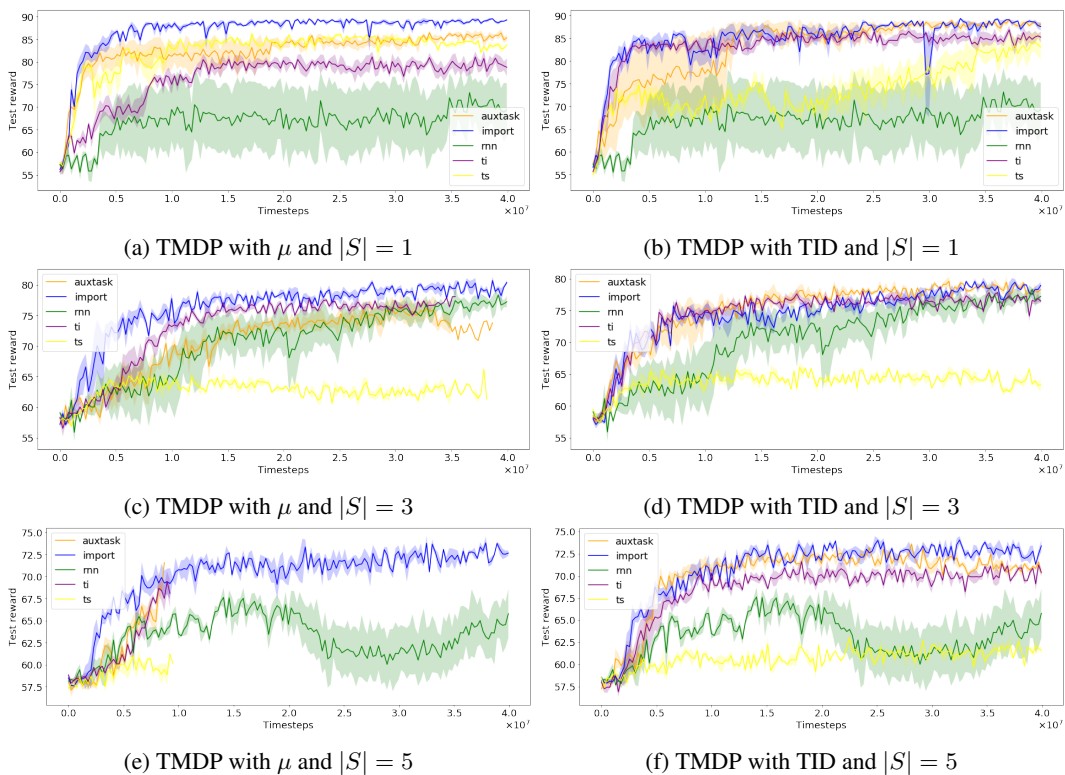

(a) TMDP with $\mu$ and $|S| = 1$

(b) TMDP with TID and $|S| = 1$

(c) TMDP with $\mu$ and $|S| = 3$

(d) TMDP with TID and $|S| = 3$

(e) TMDP with $\mu$ and $|S| = 5$

(f) TMDP with TID and $|S| = 5$

Figure 17: Evaluation on Tabular-MDP with different parameters and task descriptors (TID stands for task identifier).

# D  IMPACT OF THE $\beta$ HYPERPARAMETER

We study the sensibility of the $\beta$ parameter on IMPORT. Figure 18 clearly shows the benefits of using the auxiliary objective. On all but the Tabular-MDP environments, the recurrent policy successfully leverages the auxiliary objective to improve both sample efficiency and final performance for Acrobot.

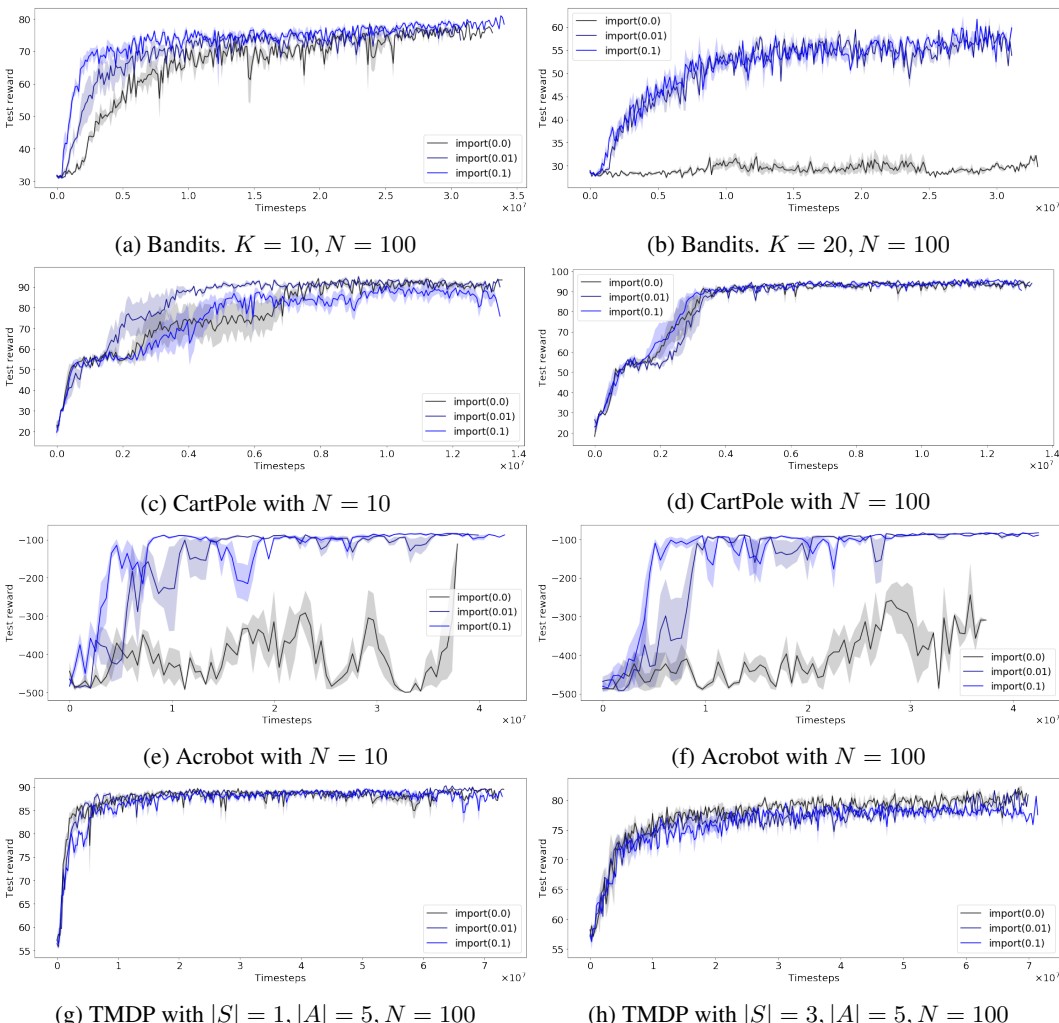

(a) Bandits. $K = 10, N = 100$

(b) Bandits. $K = 20, N = 100$

(c) CartPole with $N = 10$

(d) CartPole with $N = 100$

(e) Acrobot with $N = 10$

(f) Acrobot with $N = 100$

(g) TMDP with $|S| = 1, |A| = 5, N = 100$

(h) TMDP with $|S| = 3, |A| = 5, N = 100$

Figure 18: Test performance of IMPORT for different $\beta$ parameters (auxiliary supervised objective). We only report performance on informative $\mu$ task descriptors.

