# OpenReview forum: "Meta-Reinforcement Learning With Informed Policy Regularization"
_ICLR.cc/2021/Conference — Reject_

### Official Review · AnonReviewer1 · 2020-10-26
**Nice method, but the paper needs some more work**

**Rating:** 6
**Confidence:** 3

**Review:**

This paper presents a method that leverage task descriptors for multi-task learning. In the proposed method, an informed policy that takes in the task descriptor and the state is trained to maximize the expected return. At the same time, a RNN policy based on the history of states and actions is trained such that the RNN layers imitates the behavior of the feature extraction layers of the informed policy. In this way, the RNN policy is trained as if the task description is available. The experimental results show that the proposed method outperforms baseline methods that leverages the tasks descriptor.

The proposed method seems novel and the experimental results show its benefits. However, there are some unclear points. Especially, “online adaptation” described in the introduction is not clear.  I would like to ask the authors to clarify the following points:

- The experiment procedure is not clear to me. I understand that the informed policy and the RNN policy are trained on training tasks, but I’m not sure how the policy is adapted for test tasks. Both informed and RNN policies are further trained on test tasks?

- Does Figure 4 shows the learning curve during the training on the training tasks? If so, I recommend to add the learning curve on the test tasks to show the performance of adaptation.

- In page 7, I do not clearly understand this sentence: “Each model is trained on the training tasks, and the best model is selected on the validation tasks.” Were several models trained on training tasks? If so, how many models were trained?

- If I understand correctly, the policy is trained to maximized the expected return across the training tasks. If so, for clarify, I recommend to describe the expectation explicitly in Eq. (3), e.g, E_{\mu \sim p(\mu)} [ E_{s \sim p(s’|s, a), a \sim \pi(a|s)} [ … ] ]

- Caption of table 1 “Note that RNN does not \mu at train time.” <- something is wrong?

-	How to sample action at the initial step in the test tasks? The RNN policy seems to require the previous action a_{t-1} to generate actions, but a_{t-1} is not available in the first time step.

---

> ### Author Response · Authors · 2020-11-18
> **Author Response to Reviewer1**
>
> We thank R4 for their review.
>
> - To clarify the experimental protocol: train, test and validation tasks are sampled from the same distribution, but are not overlapping (except for the Maze3d environment since only two tasks are possible, but this environment is used to demonstrate the ability of the model to deal with large input spaces and complex exploration policies). The task information (\mu) is available only at train time. The informed policy is learned and used only at train time: a) it is used to build a relevant task embedding for training tasks that will guide the RNN to build relevant  embeddings at test and validation time b) the informed policy shares weight with the final policy, allowing the final policy to faster converge to a good solution. Again, at test and validation time, only the recurrent policy is used, the informed policy being only updated on the training environments where the task information or task identifier is known. “Online adaptation” is done thanks to the internal state of the RNN being updated at each time-step (contrary to weight updates in MAML approaches).
>
> - All curves in the paper correspond to the performance obtained on testing tasks, thus showing the ability of IMPORT to generalize to unseen tasks.
>
> - To correctly evaluate generalization to unseen environments, we use the following model selection strategy, as done in the supervised learning  paradigm:
> We train policies on training tasks, and collect their performances on validation and test tasks. To select the policies that we expect to be the best on the test tasks (i.e to avoid overfitting on the training task), we select the final policies on another set of unseen tasks (the validation tasks). Then we report the performance of these selected policies on the test tasks.
>
> - You are right. Thanks, we will describe the expectation explicitly in the next version of the paper (that we will submit before the end of the revision period)
>
> - Thanks again, the right sentence is “Note that vanilla RNN does not use \mu at train time”
>
> - In practice, actions are encoded by their one-hot encoding. At timestep = 0, the RNN is initialized with a fixed internal state (a vector of zeros), and the last action is considered to be a vector full of zeros (which does not correspond to a “real” action).
>
> An updated version of the paper will be resubmitted ASAP.

---

> > ### Comment · AnonReviewer1 · 2020-11-19
> > **question about the model selection strategy**
> >
> > I understand that multiple policies are trained on training tasks and the best one is selected based on the performance on the validation tasks. Then, I have a question about the hyperparameters of the trained policies.  Each trained model uses different set of hyperparameters? Or, multiple models are trained with the same set of hyperparameters?
> > I'm asking this because the purpose of model selection in supervised learning is to select the model with the best set of hyperparameters.  Meanwhile, in RL, the performance of the trained policies varies for different random seeds. Is the purpose of the model selection to select the best policy among the different random seeds or to select the policy trained with the best set of hyperparameters?

---

> > > ### Author Response · Authors · 2020-11-19
> > > **Author Response to Reviewer1**
> > >
> > > In our experiments, we are using task seeds to define the train, validation and test environments. Given one task seed X, the train/validation/test tasks are generated with respectively seeds X, X+100, X+1000. We use three different task seeds, i.e. X in {0,1,2} for each experiment. We use different task seeds to show that our method is robust across different sets of tasks.
> > > Now, for each task seed and method, we explore multiple hyperparameter values and follow the classical model selection schema: we select the best hyperparameter values on the validation tasks, and compute the performance of this selected policy on the test tasks.
> > >
> > > At last, for each experiment (e.g Cartpole with N=10 and task identifiers) and methog (e.g. IMPORT, AuxTask…), we report the average performance of the 3 best sets of HPs that have been selected on the 3 task seeds. Said otherwise, on the supervised learning problem, it would be like having 3 different train/test/validation datasets (e.g ImageNet, CIFAR, MNIST), doing a separate model selection over each dataset, and then reporting an average performance, which seems a realistic setting.
> > > While this selection method allows us to average performance obtained with different hyperparameter values for different task seeds, it appears that, in practice, given an experiment and a method, the value of the best hyperparameters is actually the same on all the task seeds.
> > >
> > > We will make this more clear in the next version of the paper submitted before the end of the rebuttal period.

---

> ### Author Response · Authors · 2020-11-22
> **Notification of PDF Update**
>
> We have submitted a new version of the paper where each modification is highlighted in blue. In addition, we have provided an ablation study on the auxiliary loss in the main paper. Appendix is included in the new PDF file.

---

### Official Review · AnonReviewer4 · 2020-10-29
**Simple end-to-end approach works well.**

**Rating:** 6
**Confidence:** 4

**Review:**

The authors propose an alternative architecture to handle explore/exploit tradeoffs in RL environments where each task instance may change in such a way that the policy needs to change in order to be optimal. Rather than using an explicit task inference process, and rather than relying on an RNN to slowly learn the distribution implicitly, the task id is observed during training instances and both an embedding and an RNN are trained, such that the two are interchangeable. Thus, during testing, the task id is not needed and only the RNN state is used to condition the policy. It is a straightforward way to include privileged information during training without imposing the burden of reconstruction.

The paper is clearly written and Fig 1 is very helpful to understanding the details of the architecture. The experiments are clearly explained.
The main question as a reviewer is whether the paper has significance to the community. Although it is only a small architectural contribution, the method works impressively well. It is faster to learn than Task Inference and achieves higher scores than Thompson Sampling.

It would be nice to know if the method could work in combination with other methods to quickly adapt in dynamic environments, given some labels for different features of the environments. In general, using an interchangeable embedding and RNN state is a good way to avoid the challenges of conditional architectures. The paper could be stronger if the method was framed more generally and it was shown that it could be useful on a broader range of domains that require adaptation or exploration/exploitation strategies.

---

> ### Author Response · Authors · 2020-11-18
> **Author Response to Reviewer4**
>
> Thank you for your review and comments.
>
> Concerning the significance to the community, we consider that, even if the model can be seen as an architecture change w.r.t TI or AuxTask, it is in our opinion much more than just an architectural contribution. Indeed, the main novelty of our model is to simultaneously learn an informed policy and a recurrent policy.  Training the informed policy is fast as it solves a fully-observable MDP.  The informed policy helps the discovery of the recurrent policy that efficiently manages the exploration/exploitation trade-off.  Moreover it helps in two ways:
> a) transfer to the recurrent policy through weight sharing: previous approaches did not leverage the privileged information during trajectory rollouts.
> b) providing a task embedding that is focused on the information minimally relevant for the task to solve while approaches like TI and AuxTask are based on an auxiliary supervised objective whose target can contain irrelevant or misleading information. Good learned representations enable better generalization on unseen tasks.
>
>  An updated version of the paper will be resubmitted ASAP.

---

> ### Author Response · Authors · 2020-11-22
> **Notification of PDF Update**
>
> We have submitted a new version of the paper where each modification is highlighted in blue. In addition, we have provided an ablation study on the auxiliary loss in the main paper. Appendix is included in the new PDF file.

---

### Official Review · AnonReviewer3 · 2020-10-29

**Rating:** 5
**Confidence:** 4

**Review:**

Summary: This paper studies the exploration in meta-RL problem, where a meta-RL agent must both explore (to reduce uncertainty about the task) and then solve the task. Prior RNN approaches can theoretically learn the optimal policy, but optimization can be challenging. Instead, this paper opts to leverage additional information in the form of task-descriptors that specify the task, which make the learning process easier, because the task-descriptor provides the information normally discovered via exploration. In contrast to other task-descriptor approaches, which use Thompson Sampling for exploration (and can be arbitrarily sub-optimal in certain tasks), or auxiliary losses for predicting the task-descriptor (which may try to predict task-irrelevant information), this paper instead proposes to condition on the task-descriptor as an input to the policy during meta-training. Then an RNN policy is trained in two ways: first, conditioned on the task-descriptor, which enables learning to solve the task without requiring exploration; and second, conditioned on its own hidden state, which eventually replaces the task-descriptor at meta-test time. This approach outperforms prior approaches on a suite of tasks.

Strengths:
- Clarity. Generally, this paper presents a clear and coherent narrative. The motivation for the approach is clear (better leveraging task-descriptors compared to prior approaches to more easily learn informed policies). And the approach itself is also quite understandable.
- Technical soundness. Furthermore, the approach appears to be technically sound. Leveraging the task-descriptor to learned informed policies, which are easier to optimize, can clearly produce some benefits. And then addressing the issue that the task-descriptor is unavailable at meta-test time by leveraging the hidden state also seems reasonable. The additional auxiliary loss to keep the recurrent state and task-descriptor embedding close also seems technically sound.

Weaknesses:
- Experiments. My main concern is that the experiments do not clearly substantiate the claims made in the main text:
    - The introduction of the paper claims that IMPORT “adapts faster to unknown environments, showing better generalization capabilities.” From the learning curves in Figure 4, this is not clearly the case. In 4a), 4b) and 4c), other methods seem to be learning in roughly as many samples as IMPORT requires. Notably, 4d) does show some impressive improvements, but the general claim of faster learning isn’t clearly supported by the current experiments. I was also unable to determine if the train / valid / test task splits overlapped at all, to evaluate the generalization to unknown environments part.
    - The paper also states: “We evaluate IMPORT against the main approaches to online adaptation on environments that require sophisticated exploration/exploitation strategies.” However, the environments used in the experiments, beyond the Maze3D environment don’t appear to require sophisticated exploration. I would find the experiments more compelling if IMPORT was evaluated on more complex environments requiring sophisticated exploration, like Maze3D. In a similar vein, it’s unclear to me what the takeaways from the results on the current environments should be: i.e., what does it mean about IMPORT if it performs slightly better than other approaches on CartPole or a tabular MDP? To be clear, I find evaluation on simpler environments to be useful if they clearly illustrate a point about the method, but it’s unclear to me what that point is.
    - One of the contributions of the approach, the auxiliary loss (C) is not clearly evaluated / ablated in the experiments. It appears that all of the experiments use the same value of $\beta$? But it’s not explicitly stated.
    - Finally, it would be nice if the experiments substantiate the claim that IMPORT outperforms TI by avoiding “reconstructing features in the task descriptor that are irrelevant for learning.” However, it’s not clear to me that this is occurring in the experiments. Concretely, when both approaches are given task identifiers, this problem doesn’t seem to exist, since only recovering the learning-relevant aspects is sufficient for predicting the identifier with TI. Similarly, it seems like the $\mu$’s used in the experiments contain mostly learning-relevant information.
    - In addition to the concerns about supporting the paper’s main claims, I have two additional concerns: First, the performance improvement from IMPORT is generally fairly modest. There’s only a 3-5% gain over prior approaches in CartPole, and the TabularMDP, and the results on the bandits are mixed, although the results on Maze3D are impressive.
    - Second, the TS baseline seems strangely implemented, as it’s based on “maximizing the log-likelihood of $\mu$.” It’s challenging to verify the details, because the paper states that they are in the appendix, but there is no appendix. In particular, for the bandits setting, it seems like TS should be updating a beta distribution over each arm, which I would expect to lead to stronger performance. Alternatively, it would be good to use prior approaches for TS, such as PEARL [1].

Generally, I find the proposed approach to be quite promising. This work convincingly states reasons why prior approaches (e.g., TS and task inference) sub-optimally leverage task-information. Yet, I find that the experiments insufficiently support the paper's claims, so I initially lean toward rejection.

Additional minor comments that did not affect my score:
- Related works:
    - The point about conditioning on a belief state with an RNN policy should probably cite [2].
    - Thompson Sampling as an exploration policy in meta-RL should probably cite [1].
    - [3] is also relevant to the exploration problem.
- Several key details are missing from this paper, such as the details of the environments (e.g., what is $\mu$ in the Maze3D task), which makes evaluating the experiments challenging. As mentioned above, these are reported to be in appendix, but there is no appendix.
- It would be nice to know the failure mode of other approaches on the Maze3D environment.
- The end of the setting section (Section 2) somewhat conflates the problem statement with the solution: i.e., the problem is maximizing returns, while the _proposed solution_ is “to find an architecture for $\pi$ that is able to express strategies that perform the best according to Eq. 1“
- “they approaches are sensitive” —> these approaches...

[1] Kate Rakelly, Aurick Zhou, Deirdre Quillen, Chelsea Finn, and Sergey Levine. Efficient off-policy meta-reinforcement learning via probabilistic context variables. Mar. 2019. https://arxiv.org/abs/1903.08254

[2] Zintgraf, L., Shiarlis, K., Igl, M., Schulze, S., Gal, Y., Hofmann, K., and Whiteson, S. Varibad: A very good method for bayes-adaptive deep RL via meta-learning. Oct. 2019. https://arxiv.org/abs/1910.08348

[3] Liu, E. Z.; Raghunathan, A.; Liang, P.; and Finn, C. Explore then Execute: Adapting without Rewards via Factorized Meta-Reinforcement Learning. June 2020. https://openreview.net/forum?id=La1QuucFt8-

======= UPDATE ========

I appreciate the authors' efforts during the rebuttal period, but I still retain my initial assessment of the work.

Overall, I find the proposed approach promising and easy to understand, but believe that the experiments can be improved to better substantiate the claims in this work. In particular, I believe that the benchmarks can still be more carefully chosen to better evaluate IMPORT's ability to perform sophisticated exploration. I find the 3D Maze experiment to be quite nice, as it clearly highlights a shortcoming of TS exploration, but I would find the experiments more compelling if there were additional benchmarks testing such exploration. The authors commented that exploration in meta-RL is about inferring the task to solve, which I agree with, but I think such exploration can still be made more "sophisticated" by requiring careful sequences of actions to lead to distant states, which reveal this task information.

In addition, several issues were raised regarding the TS baseline during the discussion period. The results in the bandits setting appear to be lower than those reported in other works, and it still seems like PEARL can be adapted to be a drop-in replacement for the TS baseline. I agree with the authors' assessment that the basic form of PEARL explores the setting with multiple episodes, but PEARL could just resample from the posterior every few timesteps, which is already what happens in the TS baseline.

I do think this work should be published in the future with a more careful selection of experiments.

---

> ### Author Response · Authors · 2020-11-18
> **Author Response to Reviewer3**
>
> We thank you for the constructive comments. We would like to point out that the appendix contains additional experiments and is available in the supplementary material file (the .zip file) submitted and available at the time of review.   We will update the paper to move important points of the current supplementary material  into the main paper.
>
> To be more precise on the points you raised:
>
> a. We consider the problem of generalization to new tasks, that is why the set of train/validation and test tasks are different without overlap between the tasks (except in the Maze3d experiment where only two goal locations are possible - the objective of the Maze3d experiment was not to test the generalization ability, but the capacity of the model to scale to high dimensional input spaces where the mapping between pixel inputs and \mu is complicated).
>
> Figure 4 does not show an improvement in terms of sample efficiency for the Cartpole environment, because CartPole needs a very simple exploration strategy (basically, doing random transitions is enough to identify the task to solve). We included CartPole to showcase that IMPORT’s learned representations lead to better generalization than other methods (Table 1). We would like to point out the results obtained on problems where the exploration is more complex like Bandits (Figure 13 of the appendix), tabular MDP (Figure 16) and Maze3d (Figure 15) where the improvement in terms of sample efficiency is clear. Again, these results are in the appendix for sake of space, but we will resubmit an updated version at the end of the discussion period with the additional page containing some of these curves to support our claims.
>
> b. The Bandits and Tabular-MDP (Appendix C.3 and C5) environments are two typical examples of problems where the exploration is complicated. For instance, in the Bandit environment, since IMPORT needs to address the exploration/exploitation trade-off, it needs to discover strategies like UCB, which is far from being simple to learn just from interactions. For the tabular MDP, the agent has to estimate both the reward and transition probabilities while trying to stay in high-reward states which is also a complicated strategy. Control problems (i.e CartPole and Acrobot) are mainly used to demonstrate the generalization ability of our method and we agree that they involve simpler exploration strategies.
>
> c. Following your suggestion, we performed an ablation study for the auxiliary loss. It shows that in all environments (CartPole, Acrobot, Bandits) except Tabular MDP, IMPORT benefited from the supervised auxiliary objective (in terms of sample efficiency and final performance).   The curves will be added in the appendix of the next version of the paper, which we will submit at the end of the discussion period.
>
> d. First of all, one of the ‘structural’ differences between TI and IMPORT is that TI tends to reconstruct the whole \mu information, while IMPORT is focused on the relevant part of mu discovered by learning the informed policy. Figure 4c describes an experiment where relevant information is spread over multiple features. In this case, some of the created features are almost uninformative (or redundant with other features) and the results show that IMPORT outperforms TI.
>
> e. In addition to results on Maze3d,  IMPORT  clearly outperforms all the other methods on Tabular MDP (Fig.  16  in the current version of the appendix), both in terms of sample efficiency and final performance.
>
> f. The appendix was provided in the supplementary material file together with the submission. The way we implemented TS is the following: the informed policy is used to sample episodes on which a) the log-likelihood of \mu is maximized (and computed using a RNN) b) the policy is updated to maximize the task reward.  As explained in the related work section of the paper, such training does not allow the agent to learn probing as it always acts according to an informed policy.  To circumvent this, we implemented a new version that samples from the posterior \mu distribution even at train time, yielding better results (though not matching IMPORT, AuxTask and TI’s results). We also ran TS, AuxTask and TI on bandits with a prior beta distribution: results are slightly better but still IMPORT is better when K=20. We will add those to the appendix in the next version of the paper.
>
> g. As we discuss in the comment to all reviewers, PEARL is using a different evaluation protocol.
>
> An updated version of the paper will be resubmitted ASAP.

---

> > ### Comment · AnonReviewer3 · 2020-11-19
> > **R3 Response**
> >
> > Thanks for the detailed response and for pointing out the Appendix in the supplementary, which I had missed, but have now read.
> >
> > **Generalization to new task.** The author's response addresses my concerns here.
> >
> > **Sophisticated exploration.** I agree that the bandits and tabular MDP tasks require _some_ exploration. However, from the Appendix, the largest tabular MDP considered only consists of *5 states,* and the bandits task effectively only consists of a single state. This contrasts the literature on exploration in RL, which typically considers problems with many more states, or problems that require multiple time steps of exploration to reach distant states, (e.g., the classic chain MDP example in tabular exploration).
> >
> > **TI baseline.** Figure 4c does provide compelling empirical results. It's still not clear to me why uninformative or redundant dimensions are so problematic for TI, though. If a dimension x is redundant with dimension y, then predicting x well seems like it should also predict y well. If a dimension z is uninformative, it's clear that changing the hidden state representation to make it predictive of z shouldn't help at all, but it's not clear to me why it should harm performance
> >
> > **Thompson Sampling baseline.** The implementation of this baseline still seems somewhat odd to me. Rather than directly maintaining a posterior over $\mu$'s , it seems like this baseline should maintain a posterior over a latent $z$, like PEARL. Generally, this baseline doesn't seem to type check, either. Typically, TS should involve resampling from the posterior every episode, and it's not clear when this baseline is resampling from the posterior, since there is only a single episode. With respect to the bandits task, it seems like TS in the bandits task should also update via the conjugate posterior of the beta distribution (itself another beta distribution), rather than just using a prior of the beta distribution. Using this, [1] reports significantly better results for TS in this bandits task.
> >
> > [1] https://arxiv.org/pdf/1707.02038.pdf

---

> > > ### Author Response · Authors · 2020-11-20
> > > **Author response to R3**
> > >
> > > **Sophisticated exploration.**  The exploration (probing) we are talking about is not about the exploration of the states of the MDP during the training process,  but the fact the agent needs to choose actions that allow exploring the state-space to identify the task to solve. Solving the probing/exploitation trade-off online is a POMDP problem (since mu is hidden) and is difficult even if there are only a few mu-MDP states and actions. More precisely, the difficulty of the exploration (or probing) in the Tabular MDP setting comes from the fact that the task is defined by the transition probabilities and reward over states (resulting in potentially large mu vectors, even with few states). T-MDP corresponds to a case where the probing is particularly challenging: transitions are stochastic, so in order to perform optimally, the agent needs to probe relevant state-action pairs enough times before committing (similarly to the bandits task).
> > > We will be more clear about this in the updated version.
> > >
> > > **TI baseline.**  Since TI is learned to approximate the whole mu vector (while IMPORT learns small size embeddings through the informed policy), it may take a large number of samples that will be used to approximate redundant or irrelevant mu features and reduce the sample efficiency. For instance, the RNN underlying TI may spend part of its capacity to approximate irrelevant features which is not the case with IMPORT. This is what we show in Figure 4c: when information is spread over multiple dimensions of mu, TI has difficulties to learn. It is also shown in Fig. 9 (in the appendix) where the capacity of the RNN is regulated by the embedding size: when embeddings are small, TI has more difficulties to select the relevant information than IMPORT which can rely on smaller task embeddings.
> > >
> > >
> > > **Thompson Sampling baseline.** As defined in the literature, Thompson Sampling maintains a posterior distribution on the parameters of the system it is facing (and not upon a latent variable). The originality of PEARL is to maintain such a distribution over a latent space, but this is not how Thompson Sampling is defined.  Maintaining a posterior over \mu also enables to leverage available information about \mu using a supervised objective instead of the sparse (and stochastic) reward signal. In the next version, as said previously, we will add the results on bandits using a Beta distribution as a \mu belief distribution, as described in Section 3 of [1]. Note that, if it increases the performance of TS on bandits, it still does not change our conclusions.
> > > At test time, the policy conditions on a sample from the posterior distribution: we re-sample from the posterior every k timesteps, with k an hyperparameter. As stated in Section 5, “For TS, an estimated \mu is re-sampled from the posterior every k steps, k ∈ {1, 5, 10, 20}” and k is selected through the described protocol (using validation tasks, then reported results on the test tasks).
> > >
> > > [1] Daniel Russo, Benjamin Van Roy, Abbas Kazerouni, Ian Osband, Zheng Wen. A Tutorial on Thompson Sampling. Jul. 2020.
> > >  https://arxiv.org/pdf/1707.02038.pdf

---

> > > > ### Comment · AnonReviewer3 · 2020-11-23
> > > > **R3 Response**
> > > >
> > > > Thanks for the additional clarifications and updated draft.
> > > >
> > > > To summarize, I generally like this work: the paper is clear and I find the proposed approach to be promising.
> > > >
> > > > My main concern is still about whether the claims about IMPORT learning sophisticated exploration strategies are empirically substantiated. For reference, the original RL^2 paper [1] also evaluates on the bandits task and tabular MDP task (with more states). The results may not be directly comparable between this work and [1], but [1] already shows that RL^2 outperforms UCB in the bandits task for both k = 10 and k = 50. Furthermore, RL^2 performs quite well in the tabular MDP task, although the evaluation in [1] uses 10 episodes, rather than just 1, as in this work. Consequently, regardless of whether these tasks are considered to require "sophisticated exploration," it's unclear if IMPORT is learning better exploration behaviors than RL^2, which does not even require $\mu$. This is my main hesitation for raising my score to recommend acceptance.
> > > >
> > > > [1] https://arxiv.org/abs/1611.02779

---

> > > > > ### Author Response · Authors · 2020-11-24
> > > > > **Author response to R3**
> > > > >
> > > > > We thank you for the active discussion and multiple questions. To answer to your different points:
> > > > > 1. RL^2 considers a different setting (which is different to ours and to the one in PEARL) where one trial is composed of multiple episodes: between each episode the agent returns to the first state (“episode always starts on the first state” for T-MDP in [1]). IMPORT is only evaluated on one episode (without reset) which is a different setting (and more challenging since it does not allow ‘retries’). Moreover, in our setting, we are evaluating using train/test/validation tasks, evaluating the ability of the different models to generalize, which is not the setting of RL^2.
> > > > > 2. In the bandit setting, as far as we understand, one episode is just one single step and RL^2 is exactly our RNN baseline. Moreover their bandit setting is easier than the one we propose (uniform sampling of the reward probability on the arm): since the reward probability is sampled uniformly, multiple arms provide a reasonably good reward while in our setting, only one arm provides a good amount of reward. In the first case, one just has to identify a ‘good’ arm, while our setting needs the algorithm to identify the best arm. This is the reason why all the models we have compared perform less than UCB.
> > > > > We have quickly relaunched experiments today on the MAB setting following the description provided in RL^2 (uniform sampling of the probability of the arms) for 5 and 10 arms with 100 timesteps. Our preliminary results are in line with the RL^2 paper in terms of numbers, with the RNN on par with UCB. IMPORT achieves a similar performance but seems to have a faster training. We are currently running more grid searches to finalize the results.
> > > > > 3. In addition to the differences listed above, our T-MDP configuration is also different from RL^2 in the way we sample the rewards making the performance reported in the RL^2 paper and our article not directly comparable. But note that RL^2 is close to the RNN baseline, (just having the additional reset that our baseline does not use) and we show that IMPORT is competitive over RNN.
> > > > >
> > > > > [1] https://arxiv.org/abs/1611.02779

---

> ### Author Response · Authors · 2020-11-22
> **Notification of PDF update**
>
> We have submitted a new version of the paper where each modification is highlighted in blue. In addition, we have provided an ablation study on the auxiliary loss in the main paper. Appendix is included in the new PDF file.

---

### Official Review · AnonReviewer2 · 2020-11-04
**Proposal of the advanced usage of the task descriptor for meta RL**

**Rating:** 5
**Confidence:** 4

**Review:**

Summary\
When the task descriptor is available as the privileged information, the authors propose a novel method to learn the policy that can benefit from privileged information. It is reward-driven learning and yet can make use of privileged information for efficient exploration. The advantage of the proposed method is verified in the experiments.

Comments on the paper\
I think the authors show an advantage of the proposed method by some experiments, but I‘d like to further request the following things to make the paper more convincing.

1. Because the proposed method needs the task descriptor, it would be good to explain what kind of tasks we can apply the proposed method. The wider the applicability of the proposed method is, more valuable the proposed method would be.
2. In the experiments, the authors compare with TS, TI and AuxTask. But I would like to see the comparison with another task embedding method, such as Pearl, K. Rakelly, et al., "Efficient Off-Policy Meta-Reinforcement Learning via Probabilistic Context Variables," in ICML, 2019. I guess RNN makes the training of the exploitation policy more difficult because its latent code dynamically changes especially when the state space is large such as Maze3D environment. On the other hand, the task descriptor would not change. So RNN may sometimes makes the training difficult. We can use other embedding architecture for $f_H$ such as used in Pearl. I also would like to note that when the parameter of the dynamics is used as the task descriptor, it becomes similar to Homanga Bharadhwaj et al., “MANGA: Method Agnostic Neural-policy Generalization and Adaptation”, in ICRA 2020.

----------------------------------------------------------------
Update
Thank you for the comments. But there is a misunderstanding. MANGA as well as PEARL are online methods. They just need the observed data during the episode. It can encode the observation data in online manner. I think it is not evident whether IMPORT performs better than MANGA or PERAL. I agree that RNN is general, but on the other hand, I am afraid that the internal state of RNN does not converge and usually fluctuate from time to time. It may be difficult to get a persistent policy during the episode in the same environment. I would like to encourage the authors to perform more convincing experiment and make the claim of the paper consistent with the experimental findings.

---

> ### Author Response · Authors · 2020-11-18
> **Author Response to Reviewer2**
>
> We thank you for the constructive comments.
>
> 1. Our algorithm, IMPORT,  learns representations of tasks to speed up learning for any type of task descriptors (\mu). We performed experiments with two types of task descriptors:
> a) a set of task features,
> b) the weakest possible information about the task, i.e. a task identifier / index of the train task.
> Access to such a \mu during training is a common assumption in the MTRL literature  and captures a large variety of  concrete problems:
> Type a) informative \mu at train time  is a common assumption in the literature of domain randomization [1, 2] and multi-task [3].  It is particularly relevant for problems when we have prior knowledge on physical parameters of the environment and/or the agent (robotics) or the reward function (e.g. the speed target in half cheetah).
> Type b) (See [4, 5])) mu information is less restrictive and corresponds to a large number of problems: learning in a set of N training levels in a video game, learning to drive on N different vehicles, learning to interact with N different users, learning to control N different robots, etc…
> In the two cases, IMPORT can use these tasks identifiers to generalize well. Moreover,. experimental results on  CartPole (Fig. 6) and Tabular MDP (Fig. 16) suggest that the type a) improves sample efficiency, however it does not change the final performance. Equivalent final performances of IMPORT on both types of privileged information is a desirable property and shows our method is agnostic to task descriptors.
>
> 2. As explained in the comment to all reviewers, the comparison with PEARL, MANGA, DREAM is not trivial because these models have been developed in a different setting. For instance,  MANGA uses “free” off-policy data for task identification, whereras IMPORT in an online method where the identification is made during the episode. The collected reward that we report in our paper considers the ‘price’ of the task identification, while it is ignored in the MANGA approach. Thus, MANGA does not need to address the exploration/exploitation dilemma that IMPORT addresses.
> We agree with the reviewer that using  the factored Gaussians architecture of PEARL could be an alternative to our RNN. Nonetheless, this is orthogonal to the main idea of IMPORT, and the RNN is a more general architecture with which we obtained good results, so we decided to keep it in the paper.
>
> An updated version of the paper will be resubmitted ASAP.
>
> [1] Bhairav Mehta, Manfred Diaz, Florian Golemo, Christopher J. Pal, Liam Paull. Active Domain Randomization.  Jul. 2019. https://arxiv.org/abs/1904.04762
>
> [2] Josh Tobin, Rachel Fong, Alex Ray, Jonas Schneider, Wojciech Zaremba, Pieter Abbeel. Domain Randomization for Transferring Deep Neural Networks from Simulation to the Real World. Mar. 2017. https://arxiv.org/abs/1703.06907
>
> [3] Wenhao Yu, C. Karen Liu, Greg Turk. Policy Transfer with Strategy Optimization. Dec. 2018. https://arxiv.org/abs/1810.05751
>
> [4] Jan Humplik, Alexandre Galashov, Leonard Hasenclever, Pedro A. Ortega, Yee Whye Teh, Nicolas Heess. Meta reinforcement learning as task inference. Oct 2019.  https://arxiv.org/abs/1905.06424
>
> [5] Samuel P. M. Choi, Dit-Yan Yeung, Nevin L. Zhang.  Hidden-Mode Markov Decision Processes for Nonstationary Sequential Decision Making. Dec. 2001. https://link.springer.com/chapter/10.1007/3-540-44565-X_12

---

> ### Author Response · Authors · 2020-11-22
> **Notification of PDF update**
>
> We have submitted a new version of the paper where each modification is highlighted in blue. In addition, we have provided an ablation study on the auxiliary loss in the main paper. Appendix is included in the new PDF file.

---

### Author Response · Authors · 2020-11-18
**General answer to all reviewers.**


We first thank all reviewers for their valuable comments.

A common request is to clarify our 'online adaptation setting' (R1) and compare it to PEARL/DREAM/MANGA [3,4,5] (R2, R3). In our setting, at evaluation time, a task is sampled and the performance of an algorithm is measured as the cumulative reward during a single episode. At the end of the episode, a new task is sampled. This is what we call the 'online  adaptation' setting, because the policy must explore as little as possible to maximize the reward on a single episode. This setting is standard in Meta-RL  literature [1,2], including Task Inference, which we compare to.

In contrast, PEARL, DREAM and MANGA have been developed and evaluated in a setting where at evaluation time, a task is sampled, and then 1) the learner is given a few “free”' episodes or transitions form this task (off- or on-policy, depending on the paper), followed by 2) one or several episodes. The test performance is defined as the cumulative reward collected on step 2) only. Such meta-learning settings are different from ours because stage 1) allows the policy to perform system identification while ignoring the task reward. We did not compare to these algorithms because of this discrepancy in problems addressed.

[1] Wenhao Yu, Jie Tan, C. Karen Liu, Greg Turk. Preparing for the Unknown: Learning a Universal Policy with Online System Identification. May 2017. https://arxiv.org/abs/1702.02453

[2] Jan Humplik, Alexandre Galashov, Leonard Hasenclever, Pedro A. Ortega, Yee Whye Teh, Nicolas Heess. Meta reinforcement learning as task inference. Oct 2019.  https://arxiv.org/abs/1905.06424

[3] Kate Rakelly, Aurick Zhou, Deirdre Quillen, Chelsea Finn, and Sergey Levine. Efficient off-policy meta-reinforcement learning via probabilistic context variables. Mar. 2019. https://arxiv.org/abs/1903.08254

[4] Liu, E. Z.; Raghunathan, A.; Liang, P.; and Finn, C. Explore then Execute: Adapting without Rewards via Factorized Meta-Reinforcement Learning. June 2020. https://openreview.net/forum?id=La1QuucFt8-

[5] Homanga Bharadhwaj, Shoichiro Yamaguchi, Shin-ichi Maeda. MANGA: Method Agnostic Neural-policy Generalization and Adaptation. Nov. 2019. https://arxiv.org/abs/1911.08444

---

> ### Comment · AnonReviewer3 · 2020-11-18
> **Clarification about the setting**
>
> Thanks for clarifying the setting. I have a further clarification question about the author's description of the setting:
>
> > In our setting, at evaluation time, a task is sampled and the performance of an algorithm is measured as the cumulative reward during a single episode. At the end of the episode, a new task is sampled.
>
> If I am understanding this correctly, this means that performance is evaluated on only a single episode in each test task. Is that correct?
>
> If so, how can any of the methods perform well on the bandits task? If there is only a single arm pull in the bandits task, it seems like none of the approaches have any information, and cannot do better than random. Perhaps I'm missing something? I'd love to hear the authors' thoughts.

---

> > ### Author Response · Authors · 2020-11-19
> > **Clarification about the setting to R3**
> >
> > →  Yes, this is correct. The reported reward is the reward cumulated over the whole episode, thus including reward collected while identifying the task. This  “zero-shot generalization” setting is particularly relevant for tasks where the agent is needed to perform well straight from the beginning, e.g being able to drive a new car. Note that results are reported by averaging the performance over multiple episodes and test tasks, but starting ‘from scratch’ at each new episode.
> >
> >
> > → In the bandits environment, an episode is 100 arm pulls. An arm probability vector is sampled at the beginning of the episode and remains constant during the episode.  The agent is allowed to pull an arm in [1, K] at each timestep and observes the resulting reward. It can thus identify the best arm (like UCB) by pulling them multiple times, and then pull the best arm, resulting in a policy which is more efficient than a random one. This is what IMPORT (and other baselines) is doing: discovering an efficient exploration policy that does not ‘consume’ too much reward.
> >
> > Our evaluation setting is a perfect fit for environments as bandits as the trade-off between probing and exploitation is of utmost importance. The setting from PEARL, DREAM, MANGA would make bandits trivial following a random policy to gain information during the free episode then exploiting the inferred best arm during other episodes.

---

### Decision · Program_Chairs · 2021-01-07
**Final Decision**

**Decision:**

Reject

**Comment:**

This paper is borderline, as evidenced by all of the reviewer's scores.

The pros are:
- important and relevant topic
-  IMPORT is a reasonable, technically sound approach
- paper is relatively clear

The cons all lie in the experimental evaluation, and whether the experiments sufficiently back the claim that IMPORT can learn sophisticated exploration strategies and validate IMPORT's merits compared to prior algorithms. In particular:
- The choice of benchmarks does not sufficiently test the ability to explore in a sophisticated manner
- Lack of comparisons to PEARL and MANGA, which can readily be applied to the online setting
- The empirical improvements are relatively modest.

Overall, the cons slightly outweigh the pros of the paper. Indeed, no reviewer was willing to champion the paper's acceptance.